# Learning the RoPEs: Better 2D and 3D Position Encodings with STRING

**Connor Schenck** [* 1 †]  **Isaac Reid** [* 2 3 † ^]  **Mithun George Jacob** [* 1 †]  **Alex Bewley** [* 1 †]  **Joshua Ainslie** [* 1 †]
**David Rendleman** [* 1 †]  **Deepali Jain** [* 1]  **Mohit Sharma** [* 1]  **Avinava Dubey** [* 3]  **Ayzaan Wahid** [1]  **Sumeet Singh** [1]
**René Wagner** [1]  **Tianli Ding** [1]  **Chuyuan Fu** [1]  **Arunkumar Byravan** [1]  **Jake Varley** [1]  **Alexey Gritsenko** [1]
**Matthias Minderer** [1]  **Dmitry Kalashnikov** [1]  **Jonathan Tompson** [1]  **Vikas Sindhwani** [1]
**Krzysztof Choromanski** [* 1 ‡]

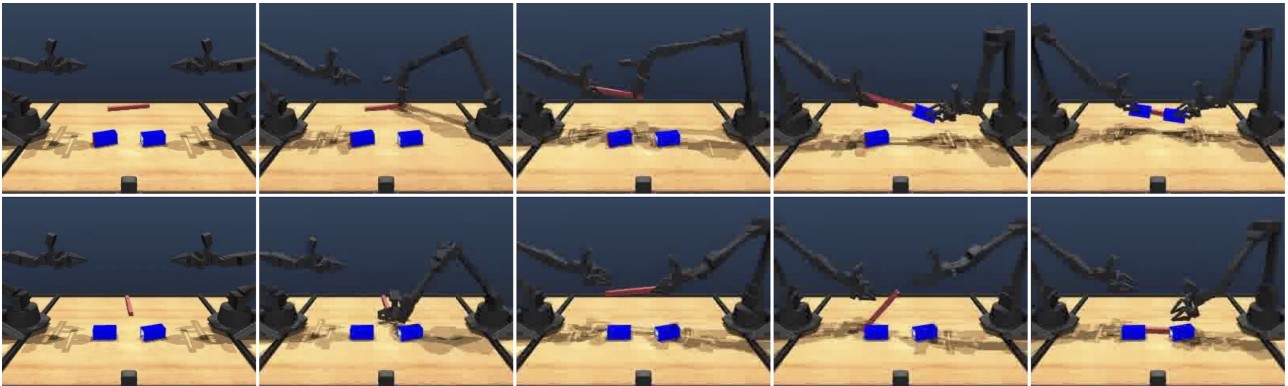

Figure 1. **Top:** Successful diffusion policy conditioned on a STRING-enhanced Transformer vision encoder, attempting the double-insertion task on Aloha-sim. **Bottom:** Same experiment, but with a regular vision encoder for which the policy fails. STRING provides strong improvements for training dexterous robotics policies, outperforming previous position encoding algorithms such as RoPE.

## Abstract

We introduce **STRING**: Separable Translation-ally Invariant Position Encodings. STRING extends Rotary Position Encodings (RoPE; Su et al., 2024), a recently proposed and widely used algorithm in large language models, via a unifying theoretical framework. Importantly, STRING still provides **exact** translation invariance, including token coordinates of arbitrary dimensionality, whilst maintaining a low computational footprint. These properties are especially important in robotics, where efficient 3D token representation is key. We integrate STRING into Vision Transformers with RGB(-D) inputs (color plus optional depth), showing substantial gains, e.g. in open-vocabulary object detection and for robotics controllers. We complement our experiments with a rigorous mathematical analysis, proving the universality of our methods. Videos of STRING-based robotics controllers can be found here.

---

[*]Equal contribution [†]Random order [^]Work done at Google Research [‡]Senior lead. [1]Google DeepMind [2]University of Cambridge [3]Google Research. Correspondence to: Krzysztof Choromanski <kchoro@google.com>.

*Proceedings of the 42nd International Conference on Machine Learning*, Vancouver, Canada. PMLR 267, 2025. Copyright 2025 by the author(s).

## 1. Introduction and Related Work

Position encodings (PEs) (Zhao et al., 2023; Kazemnejad et al., 2023; Chen et al., 2021; Kiyono et al., 2021) inject information about the respective locations of tokens into transformers (Vaswani et al., 2017). They are essential for good performance because vanilla attention is a set function, equivariant under permutation. In contrast, the meaning of a sequence of tokens in general depends on its ordering.

**APEs and RPEs.** Practitioners initially relied on *absolute* PEs (APEs; Vaswani et al., 2017; Kiyono et al., 2021; Wang et al., 2021; Liu et al., 2020) which add or concatenate fixed, precomputed position embeddings to tokens. These have since been replaced by *relative* PEs (RPEs; Shaw et al., 2018; Raffel et al., 2020; Li et al., 2024; Chi et al., 2022; Press et al., 2022; Chi et al., 2023), which add a learnable bias term that depends on the distance between tokens to the pre-softmax attention logits. RPEs tend to generalise better than APEs over varying sequence lengths. However, they often require explicit computation for every query-key pair.

**RoPE.** To address the limitations of RPEs and APEs, researchers recently introduced *rotary* position encodings (RoPE; Su et al., 2024; Heo et al., 2025). These have been widely adopted in large language models (LLMs; Dubey et al., 2024; Gemma Team et al., 2024). RoPE acts on queries and keys by partitioning them into 2-dimensional blocks, each of which is rotated by an angle proportional

to the token's position in the sequence. Whilst queries and keys are rotated separately, the angle of *relative* rotation is proportional to their separation, combining the best properties of APEs and RPEs. Mathematically, for query and key of dimensionality $d$, RoPE involves $\lfloor \frac{d}{2} \rfloor$ Givens rotations (Bindel et al., 2002) acting on disjoint 2D subspaces.

Besides providing strong empirical gains, two attractive properties have driven the enthusiastic uptake of RoPE.

1. **Separability.** RoPE transforms each query and key independently, based on its position. This happens once per token; the PE'd tokens are not recalculated during subsequent processing like autoregressive generation. This makes KV-caching convenient. Separability also makes RoPE compatible with linear attention, e.g. Performers (Choromanski et al., 2020; Katharopoulos et al., 2020). Here, the attention matrix is not instantiated in memory so explicit RPE mechanisms are not possible.[1]

2. **Translational invariance.** For a query-key pair at positions $(i, j) \in \mathbb{N}^2$, the relative rotation angle depends only on $i - j$. This improves sequence-length generalization.

However, RoPE is not the only position encoding algorithm with these desirable traits. In this paper, we propose a more general algorithm called **STRING**: **S**eparable **Tr**anslationally **In**variant Position Encodin**g**s. STRING is based on Lie groups. It generalises RoPE via a unifying theoretical framework, incorporating the latter as a special case. In fact, we later prove that STRING is the *most general* PE algorithm with the properties above, amongst a broad class.

**STRING for robotics.** The above features are especially important in robotics, where efficient 2D/3D token representation and sensible physical priors are key. To demonstrate it, we integrate STRING into Vision Transformers (ViTs), showing strong improvements for open-vocabulary object detection models and various robotics controllers. This showcases the real-world impact of our STRING.

Videos of STRING-based robotics controllers can be found here: https://sites.google.com/view/string-robotics.

**Key contributions.**

1. We introduce STRING, a new family of position encodings for multidimensional token coordinates that respect both separability and translational invariance.

2. We rigorously analyse STRING's theoretical properties (**Sec. 3**), proving that it is more general than RoPE. We provide computationally efficient implementations.

3. We show strong accuracy gains across varied models using Transformers with STRING, on a range of robotics and general vision tasks (see Fig. 1 and **Sec. 4**).

---

[1]*Implicit* relative position encoding schemes, which avoid instantiating the attention matrix in memory, have also been proposed (Reid et al., 2024; Choromanski et al., 2022; Luo et al., 2021).

## 2. Preliminaries

Let $\{\mathbf{x}_i\}_{i=1}^N \in \mathbb{R}^d$ denote a set of $N$ $d$-dimensional tokens. Assume that $d$ is even. The $i$th query, key and value vectors are given by $\mathbf{q}_i = \mathbf{W}_q \mathbf{x}_i$, $\mathbf{k}_i = \mathbf{W}_k \mathbf{x}_i$ and $\mathbf{v}_i = \mathbf{W}_v \mathbf{x}_i$ respectively, where $\mathbf{W}_q, \mathbf{W}_k, \mathbf{W}_v \in \mathbb{R}^{d \times d}$ are learned projection matrices (to keep the notation simple, we assume here the one-head setting). The *attention mechanism*, basic computational unit of the Transformer, can be written as:

$$\mathbf{x}_i \to \frac{\sum_j \exp(\mathbf{q}_i^\top \mathbf{k}_j) \mathbf{v}_j}{\sum_l \exp(\mathbf{q}_i^\top \mathbf{k}_l)}. \tag{1}$$

This updates the set of tokens, mixing them dynamically depending on the query-key softmax similarity scores.

**Rotary position encodings.** As described in Section 1, RoPE rotates tokens depending on their locations. In the 1D data setting (e.g. text), for a token $\mathbf{z}_i \in \{\mathbf{q}_i, \mathbf{k}_i\}$ at position $i \in \mathbb{N}$, we take $\mathbf{z}_i \to \text{RoPE}(i)\mathbf{z}_i$ with

$$\text{RoPE}(i)\mathbf{z}_i \coloneqq \bigoplus_{n=1}^{d/2} \boldsymbol{\rho}(i\theta_n)[\mathbf{z}_i]_{2n-1:2n}, \tag{2}$$

$$\boldsymbol{\rho}(\theta) \coloneqq \begin{bmatrix} \cos\theta & -\sin\theta \\ \sin\theta & \cos\theta \end{bmatrix}. \tag{3}$$

Here, $\bigoplus$ denotes the direct product, so that each $2 \times 2$ matrix $\{\boldsymbol{\rho}(i\theta_n)\}_{i=1}^{d/2}$ independently rotates a 2-element section of the query/key, and $[\mathbf{z}_i]_{2n-1:2n}$ denotes the $2n - 1$ and $2n$ elements of $\mathbf{z}_i$. Note that the matrix $\text{RoPE}(i)$ is $d \times d$, but it is only nonzero on the $2 \times 2$ blocks on the diagonal. Since $\boldsymbol{\rho}(\theta)^\top = \boldsymbol{\rho}(-\theta)$ and 2D rotations commute, we have that

$$\text{RoPE}(i)^\top \text{RoPE}(j) = \text{RoPE}(j - i), \tag{4}$$

whereupon we are transforming $\mathbf{q}_i^\top \mathbf{k}_j \to \mathbf{q}_i^\top \text{RoPE}(j-i)\mathbf{k}_j$. The dependence on $j - i$ makes this translationally invariant. RoPE takes the set of angles $\{\theta_n\}_{n=1}^{d/2}$, determining the rotation frequency of each $2 \times 2$ block, as hyperparameters. We suppress this dependence for notational compactness. The authors originally proposed the decaying sequence $\theta_n = \lambda^{-2(n-1)/d}$ with base wavelength $\lambda = 10,000$, though variants have since been explored (see below).

**RoPE in higher dimensions.** Whilst RoPE was originally proposed for sequence data, recent work has extended it to encode higher-dimensional position information (Heo et al., 2025). Now each token is equipped with a vector $\mathbf{r}_i \in \mathbb{R}^{d_c}$, and we require: $\text{RoPE}(\mathbf{r}_i)^\top \text{RoPE}(\mathbf{r}_j) = \text{RoPE}(\mathbf{r}_j - \mathbf{r}_i)$. Since 2D rotations commute, one approach is to define

$$\text{RoPE}(\mathbf{r}_i) \coloneqq \prod_{k=1}^{d_c} \text{RoPE}([\mathbf{r}_i]_k), \tag{5}$$

where $[\mathbf{r}_i]_k$ is the $k$th coordinate of $\mathbf{r}_i$ (with $k \in \{1, ..., d_c\}$). This independently applies regular 1-dimensional RoPE (Eq. 2) for each dimension of the position vector. The

rotation frequencies $\{\theta_n\}_{n=1}^{d/2}$ can optionally differ between each coordinate axis.

**Generalisations of RoPE.** Prompted by its success, a number of papers have since sought to understand the effectiveness of RoPE and propose better-performing alternatives. One well-known method argues to increase the base wavelength $\lambda$ to $500,000$, slowing the rate of token rotation and improving learning with longer contexts (Xiong et al., 2023; Roziere et al., 2023). Another suggests to completely truncate the lowest frequencies, setting them to zero, which helps preserve long-range 'semantic channels' (Barbero et al., 2024). Practitioners can also make the parameters $\{\theta_n\}_{n=1}^{d/2}$ fully learnable, improving flexibility. Lastly, recent work has proposed to replace the block-diagonal RoPE matrices $\text{RoPE}(i)$ by more general dense matrices in $\text{SO}(d)$, parameterized by learned antisymmetric generators (Ostmeier et al., 2024). Whilst more expressive, this algorithm breaks translational invariance for position vectors with $d_c > 1$, and has a large memory footprint. This makes it unsuitable for robotics applications. In Section 3, we will propose a better alternative, STRING.

# 3. STRING: Separable Translationally Invariant Position Encodings

Recall that our goal is modify queries and keys depending on their respective positions, so that changes to dot products $\mathbf{q}_i^\top \mathbf{k}_j$ depend on $\mathbf{r}_i - \mathbf{r}_j$. RoPE achieves this using matrix multiplication (Su et al., 2024). Here, we present STRING: a more general, better-performing algorithm.

## 3.1. Introducing STRING

STRING is defined as follows.

---

**Definition 3.1.** STRING is the mapping $\mathbf{R}(\cdot)$ : $\mathbb{R}^{d_c} \to \mathbb{R}^{d \times d}$, from $d_c$-dimensional position vectors to $d \times d$ matrices, given by

$$\mathbf{R}(\mathbf{r}_i) = \exp\left(\sum_{k=1}^{d_c} \mathbf{L}_k [\mathbf{r}_i]_k\right), \quad (6)$$

where $\{\mathbf{L}_k\}_{k=1}^{d_c} \subset \mathbb{R}^{d \times d}$ is a set of learnable and commuting skew-symmetric generators. Given a set of queries or keys $\{\mathbf{z}_i\}_{i=1}^N \subset \mathbb{R}^d$ at positions $\{\mathbf{r}_i\}_{i=1}^N \subset \mathbb{R}^{d_c}$, their positions are encoded as:

$$\mathbf{z}_i \to \mathbf{R}(\mathbf{r}_i)\mathbf{z}_i \quad \forall i \in \{1, ..., N\}. \quad (7)$$

---

Here, $\exp(\cdot)$ refers to the *matrix* exponential, defined by its series expansion $\exp(\mathbf{A}) := \sum_{i=0}^{\infty} \mathbf{A}^i/i!$ and $[\mathbf{r}_i]_k$ is the $k$th coordinate of vector $\mathbf{r}_i$. By 'commuting skew-symmetric generators', we mean that $\{\mathbf{L}_k\}_{k=1}^{d_c}$ satisfy

$$[\mathbf{L}_i, \mathbf{L}_j] = 0 \quad \text{and} \quad \mathbf{L}_i^\top = -\mathbf{L}_i \ \forall \ i, j. \quad (8)$$

There are many ways to parameterize such a set; we give examples in Section 3.2. Remarkably, the following is true.

**Theorem 3.2** (STRING is general). *Consider the set of mappings $\mathbf{R}(\cdot) : \mathbb{R}^{d_c} \to \mathbb{R}^{d \times d}$ that satisfy the group-like translational invariance property $\mathbf{R}(\mathbf{r}_i)^\top \mathbf{R}(\mathbf{r}_j) = \mathbf{R}(\mathbf{r}_j - \mathbf{r}_i) \forall \mathbf{r}_i, \mathbf{r}_j \in \mathbb{R}^{d_c}$, are continuously differentiable with respect to $\mathbf{r}_i$, and satisfy $\mathbf{R}(\mathbf{0}) = \mathbf{I}_d$ (with $\mathbf{I}_d$ the $d$-dimensional identity). All such mappings can be expressed as STRING with some set of generators $\{\mathbf{L}_k\}_{k=1}^{d_c} \subset \mathbb{R}^{d \times d}$.*

In this sense, STRING is the *most general* of all translationally invariant position encoding mechanisms using matrix multiplication. Meanwhile, RoPE is a simple special case of STRING, taking a particular choice of generators. This can be seen as follows.

**Theorem 3.3** (RoPE is a type of STRING #1). *Consider the generators $\mathbf{L}_k = \sum_{p=1}^{d/2}(\delta_{2p,2p-1} - \delta_{2p-1,2p})\theta_p$, where $\{\theta_p\}_{p=1}^{d/2} \subset \mathbb{R}$ and $\delta_{i,j}$ is the delta function. This corresponds to RoPE with rotational frequencies $\{\theta_p\}_{p=1}^{d/2}$.*

Proofs of Theorem 3.2 and Theorem 3.3 are in Appendix A.

## 3.2. Computationally efficient STRING

Despite being general and notationally compact, the parameterization of the STRING matrices $\mathbf{R}(\mathbf{r}_i)$ shown in Eq. 6 may not be convenient for practical applications. Given $N$ tokens at positions $\{\mathbf{r}_i\}_{i=1}^N$, one must in general exponentiate and store $N$ dense $d \times d$ matrices. This incurs $\mathcal{O}(Nd^3)$ time complexity and $\mathcal{O}(Nd^2)$ space complexity cost. The problem is exacerbated if the $\{\mathbf{r}_i\}_{i=1}^N$ differ across training examples and batches, which occurs e.g for point cloud data or color plus depth channel (RGB-D) images. In this case, $\{\mathbf{R}(\mathbf{r}_i)\}_{i=1}^N$ cannot be cached and reused. This motivates the goal of this section: to find *efficient* STRING instantiations, nonetheless more general and expressive than RoPE. We begin with the following (proof in Appendix A):

**Theorem 3.4** (RoPE is a type of STRING #2). *For any STRING position encoding with generators $\{\mathbf{L}_k\}_{k=1}^{d_c}$, there exists an orthogonal matrix $\mathbf{P}$ such that*

$$\mathbf{R}(\boldsymbol{r}_i) = \mathbf{P}\text{RoPE}(\boldsymbol{r}_i)\mathbf{P}^\top. \quad (9)$$

Note that the orthogonal matrix $\mathbf{P}$ is independent of the coordinates $\mathbf{r}_i$, so it can be learned and stored once per attention head and shared across all training examples. Meanwhile, $\text{RoPE}(\boldsymbol{r}_i)$ is sparse – it is only nonzero on the super- and subdiagonals – so multiplying tokens only requires $\mathcal{O}(Nd)$ memory and $\mathcal{O}(Nd^2)$ time, saving a factor of $d$. This is crucial in the contrastive learning setting where batch sizes can become large. Once again, one can see that RoPE is a special case of STRING, this time taking $\mathbf{P} = \mathbf{I}_d$. We emphasize that the parameterization of STRING in Eq. 9 remains just as general as in Eq. 6. We also note that, since

in regular attention one takes dot products between position-encoded queries and keys, the first orthogonal matrix $\mathbf{P}$ will always cancel with its counterpart. Therefore, in Transformers it is sufficient to take $\mathbf{R}(\boldsymbol{r}_i) = \mathrm{RoPE}(\boldsymbol{r}_i)\mathbf{P}$, with $\mathbf{P} \in \mathrm{O}(d)$ learnable, without loss of generality.[2]

**Example 1: Cayley-STRING.** Equipped with Theorem 3.4, our goal becomes to choose a suitable parameterization for the orthogonal matrix $\mathbf{P}$. One option is to take the *Cayley Transform*,

$$\mathbf{P}_{\mathrm{Cayley}} := (\mathbf{I}_d - \mathbf{S})(\mathbf{I}_d + \mathbf{S})^{-1}, \qquad (10)$$

where $\mathbf{S}$ is a learnable (potentially sparse) antisymmetric matrix (Diele et al., 1998). $\mathbf{P}_{\mathrm{Cayley}}$ is convenient since, for a token $\mathbf{z}_i$, we can compute $(\mathbf{I}_d + \mathbf{S})^{-1}\mathbf{z}_i$ efficiently using a linear solver, avoiding expensive methods such as matrix inversions and matrix exponentials. Note that wherever we use $\mathbf{P}_{\mathrm{Cayley}}$, we refer to our algorithm as *Cayley-STRING*.

**The unreasonable effectiveness of STRING.** In some sense, Theorem 3.4 makes it surprising that STRING outperforms RoPE so comprehensively in all our experiments (see Section 4), given that they are related by a change of basis. It appears that the ability to *explicitly* learn this basis change via $\mathbf{P}$ (underline{shared} between queries and keys), rather than implicitly via existing network weights, substantially boosts performance. Conversely, when using linear attention variants, the projected tokens $\mathbf{W}_q\mathbf{q}_i$ and $\mathbf{W}_k\mathbf{k}_i$ are pushed through nonlinearities such as $\mathrm{ReLU}(\cdot)$ before taking the dot product. Hence, in this case, including learnable $\mathbf{P}$ does increase the capacity of the network, rather than simply learning a basis change.

**Example 2: Circulant-STRING.** We now present a second efficient STRING algorithm within our framework. A square matrix is referred to as *circulant* if it takes the form

$$\mathbf{C} = \begin{bmatrix} c_0 & c_{d-1} & \cdots & c_2 & c_1 \\ c_1 & c_0 & c_{d-1} & \cdots & c_2 \\ \vdots & c_1 & c_0 & \ddots & \vdots \\ c_{d-2} & \vdots & \ddots & \ddots & c_{d-1} \\ c_{d-1} & c_{d-2} & \cdots & c_1 & c_0 \end{bmatrix}. \qquad (11)$$

All rows are composed of the same elements, and each row is rotated one element relative to the preceding row. The transpose of a circulant matrix $\mathbf{C}^\top$ is also circulant, and the sum of two circulant matrices is also circulant. It follows that $\mathbf{C} - \mathbf{C}^\top$ is circulant and antisymmetric. Lastly, circulant matrices commute. With these properties in mind, we can simply define $\mathbf{L}_k = \mathbf{C}_k - \mathbf{C}_k^\top$ for $k \in \{1, ..., d_c\}$, with $\mathbf{C}_k$ a learnable circulant matrix parameterized by $d$ scalars $\{c_0, ..., c_{d-1}\}$. We call this *Circulant-STRING*. This special parameterization is convenient for the following reason.

---

[2]We dropped the transpose sign on the second $\mathbf{P}$, redefining $\mathbf{P}^\top$ as our learnable orthogonal matrix.

**Theorem 3.5** (Circulant-STRING is fast). *Given generators* $\mathbf{L}_k = \mathbf{C}_k - \mathbf{C}_k^\top$ *with* $\mathbf{C}_k$ *circulant, the position encoding* $\exp(\sum \mathbf{L}_k[\mathbf{r}_i]_k)\mathbf{z}_i$ *for token* $\mathbf{z}_i$ *at position* $\mathbf{r}_i$ *can be computed in* $\mathcal{O}(d\log d)$ *time and* $\mathcal{O}(d)$ *memory using the fast Fourier Transform (FFT).*

We provide a proof in Appendix A. Circulant-STRING provides another efficient position encoding algorithm that scales gracefully to large, high-dimensional datasets and performs well in spatial applications (see Section 4).

**Learnable frequencies with STRING.** Note that the STRING generators from Definition 3.1 are (in general) learnable. For Cayley-STRING, the angle-defining frequencies for RoPE and $\mathbf{S}$, the skew-symmetric matrix from Equation (10) are learned whereas in Circulant-STRING, the scalars $\{c_0, ..., c_{d-1}\}$ in Equation (11) are learned.

| **STRING** | **Train** | | **Inference** | |
|---|---|---|---|---|
| | Space | Time | Space | Time |
| Cayley | $\mathcal{O}(d^2)$ | $\mathcal{O}(d^3)$ | $\mathcal{O}(d)$ | |
| Circulant | $\mathcal{O}(d)$ | $\mathcal{O}(d\log d)$ | $\mathcal{O}(d)$ | $\mathcal{O}(d\log d)$ |

*Table 1.* Space and time complexity of the presented STRING methods in terms of the token dimensionality per head $d$.

**Computational complexity.** Table 1 lists the computational complexity of the presented STRING methods. With token dimensionality per head $d$, Cayley-STRING introduces $d^2/2$ parameters per head (from the $d/2$ angle-defining RoPE frequencies and $d(d-1)/2$ parameters which fully determine the learnable skew-symmetric matrix used to generate $\mathbf{P}_{\mathrm{Cayley}}$). Circulant-STRING introduces $d$ parameters per head (from the learnable circulant matrix fully determined by $d$ parameters). In our experiments, $d = 64$ resulted in a negligible increase of trainable parameters.

During training, Cayley-STRING takes $\mathcal{O}(d^3)$ time due to the linear solver. However, during inference, the learned orthogonal matrix $\mathbf{P}_{\mathrm{Cayley}}$ can be absorbed into existing $q/k$ projections at no extra cost (for vanilla attention). Circulant-STRING only takes $\mathcal{O}(d\log d)$ time via the Fast Fourier Transform.

Both approaches improve upon RoPE, yet Cayley-STRING in general leads to larger improvements (see Section 4). Thus, we have here a classic trade-off between quality and computational expense. For applications with strict training performance constraints, we recommend Circulant-STRING due to its compact computational footprint, whereas for other applications Cayley-STRING is recommended.

### 3.3. Loose ends

Here, we discuss further generalisations of STRING.

**Extension 1: ⊗-STRING.** So far, we have followed RoPE in assuming that our position encodings are applied via matrix multiplication. However, this can be relaxed whilst preserving separability and translational invariance. For example, one can transform tokens $\mathbf{z}_i$ via the *outer* product with position feature vectors $\mathbf{f}(\mathbf{r}_i) \in \mathbb{R}^{2m}$,

$$\mathbf{z}_i \rightarrow \text{vec}(\mathbf{f}(\mathbf{r}_i) \otimes \mathbf{z}_i). \tag{12}$$

Here, $\otimes$ denotes the outer product and vec denotes the 'vectorizing' operation that flattens a matrix to a vector, so that $\text{vec}(\mathbf{f}(\mathbf{r}_i) \otimes \mathbf{q}_i)_{da+b} = \mathbf{f}(\mathbf{r}_i)_a \mathbf{q}_{ib}$ where $a \in \{1, ..., 2m\}$ and $b \in \{1, ..., d\}$. Since the dot product of (flattened) outer products gives the product of dot products, we have

$$\text{vec}(\mathbf{f}(\mathbf{r}_i) \otimes \mathbf{q}_i)^\top \text{vec}(\mathbf{f}(\mathbf{r}_j) \otimes \mathbf{k}_j) = \mathbf{q}_i^\top \mathbf{k}_j \cdot \mathbf{f}(\mathbf{r}_i)^\top \mathbf{f}(\mathbf{r}_j). \tag{13}$$

Now suppose that we take the Fourier features

$$\mathbf{f}(\mathbf{r}_i) = \frac{1}{\sqrt{m}} \left[ \cos(\boldsymbol{\omega}_k^\top \mathbf{r}_i), \sin(\boldsymbol{\omega}_k^\top \mathbf{r}_i) \right]_{k=1}^m, \tag{14}$$

where $\{\boldsymbol{\omega}_k\}_{k=1}^m \subset \mathbb{R}^d$ are learnable $d$-dimensional frequency vectors. Then we have that $\mathbf{f}(\mathbf{r}_i)^\top \mathbf{f}(\mathbf{r}_r) = \frac{1}{m} \sum_{k=1}^m \cos(\boldsymbol{\omega}_k(\mathbf{r}_i - \mathbf{r}_j))$ which is clearly a function of $\mathbf{r}_i - \mathbf{r}_j$. We refer to this novel position encoding variant, orthogonal to previous RoPE-like approaches, as ⊗-*STRING*.

**Extension 2: General transformation groups.** Having focused on *translational* invariance, another natural question is whether STRING could be repurposed for other continuous transformation groups. These may be more suitable for data with different properties; for example, one might sometimes prefer a *rotationally* invariant position encoding.

More formally, recall that a Lie group with parameters $\psi \in \mathbb{R}^k$ is a group of transformations of the form $T_\psi : \mathbb{R}^d \rightarrow \mathbb{R}^d$ that are differentiable with respect to $\psi$. Let the parameter $\psi = 0$ correspond to the identity element, so that $T_0\mathbf{x} = \mathbf{x}$. A *canonical coordinate system* for $G$ is an injective map $\rho$ from Cartesian coordinates to a new coordinate system, satisfying $\rho(T_\psi\mathbf{x}) = \rho(\mathbf{x}) + \sum_{i=1}^k \psi_i \mathbf{e}_i \, \forall \, T_\psi \in G$, where $\mathbf{e}_i$ is the $i$th standard basis vector. Observe that the right hand side of this equation represents a translation in the new basis. Canonical coordinate systems exist for all one-parameter Lie groups ($k = 1$), and more generally for Abelian groups of dimension $k \leq d$ (Segman et al., 1992; Rubinstein et al., 1991; Tai et al., 2019). They can be derived analytically by solving a set of first-order PDEs, though for many common transformation groups the canonical coordinate system is obvious. For instance, for azimuthal and polar rotations of points $(\mathbf{r}_x, \mathbf{r}_y, \mathbf{r}_z)$ in 3D space ($k = 2$), a canonical coordinate system is $(\theta, \phi)$, where $\sin\theta := \sqrt{\mathbf{r}_x^2 + \mathbf{r}_y^2}/\|\mathbf{r}\|_2$ and $\tan\phi := \mathbf{r}_y/\mathbf{r}_x$. Rotating[3] simply 'translates' the canonical

---

[3] Note that this differs from full 3D object pose invariance, for which the transformations do *not* form an Abelian group.

---

coordinates $(\theta, \phi) \rightarrow (\theta + \Delta\theta, \phi + \Delta\phi)$ – a transformation looking much more complicated in the Cartesian basis.

**STRING for Abelian Lie groups.** It follows that, simply by replacing Cartesian coordinates $\{\mathbf{r}_i\}_{i=1}^N$ with their canonical counterparts, we can repurpose STRING to construct position encodings that respect more general invariances.

## 4. Experiments

In this section, we provide an exhaustive empirical comparison of STRING with RoPE and vision encoders leveraging regular APEs. To set the stage, we start with general non-robotics experiments in Sec. 4.1. On our way to robotics applications, we then test STRING for 2D and 3D object detection in Sec. 4.2. Finally, we present robotics manipulation experiments in Sec. 4.3 and Sec. 4.4.

### 4.1. General Experiments: Classification and Retrieval

We tested STRING for image classification tasks on the ImageNet2012 (Deng et al., 2009) and Places365 datasets, with Vision Transformer (ViT) (Dosovitskiy et al., 2021) as our base model. We compare against RoPE and RoPE-Mixed (Heo et al., 2025), abbreviated to RoPE-M, to Circulant-STRING and Cayley-STRING (respectively abbreviated to Circulant-S and Cayley-S). The results are shown in Table 2. For both datasets, STRING offers best models. For ImageNet2012, the top two models are STRINGs. Furthermore, ImageNet2012 STRINGs provide absolute gains larger than $1\%$, as compared to regular ViTs, with only a negligible set of extra trainable parameters.

| | ViT | RoPE | RoPE-M | Circulant-S | Cayley-S |
|---|---|---|---|---|---|
| ImageNet | 80.04 | 80.18 | 80.86 | **81.22** | 81.09 |
| Places365 | 56.79 | 56.97 | 56.69 | 56.77 | **57.16** |
| Mean | 68.42 | 68.58 | 68.78 | 69.00 | **69.12** |

*Table 2.* Image classification % test accuracy. Best numbers are highlighted in **bold** and the second-best numbers are underlined.

| | i2t@1 | i2t@5 | i2t@10 | t2i@1 | t2i@5 | t2i@10 | Mean |
|---|---|---|---|---|---|---|---|
| ViT | 53.88 | 73.17 | 78.49 | 53.98 | 73.83 | 79.29 | 68.77 |
| RoPE | 55.27 | 74.27 | 79.52 | 55.22 | 74.61 | 80.25 | 69.86 |
| RoPE-M | 55.30 | 74.08 | 79.47 | 55.36 | 74.73 | 80.18 | 69.85 |
| Circulant-S | 55.52 | 74.69 | 79.91 | 55.68 | 75.03 | 80.45 | 70.21 |
| Cayley-S | **55.70** | **75.08** | **80.24** | **55.82** | **75.40** | **80.65** | **70.48** |

*Table 3.* Image-to-text (i2t) and text-to-image (t2i) WebLI recall @ rank (best numbers: in **bold**, second-best: underlined.)

### 4.2. Improving Open-Vocabulary Object Detection

#### 4.2.1. OPEN-VOCABULARY OBJECT DETECTION IN 2D

Next, we lift WebLI (Chen et al., 2023), a dataset of 10 billion image-text pairs across a variety of languages, into 3D by pre-processing a 60-million image subset with Depth-

|  | Baseline | RoPE | RoPE-M | Circulant-S | Cayley-S |
|---|---|---|---|---|---|
| $AP^{COCO}$ | 32.44 | **33.66** | 32.74 | 33.24 | 33.47 |
| $AP^{LVIS}$ | 21.98 | 22.71 | 22.43 | 22.60 | **23.01** |
| Mean | 27.21 | 28.18 | 27.59 | 27.92 | **28.24** |

*Table 4.* Average Precision (AP) % of the OWL-ViT model on COCO (Lin et al., 2014) and LVIS (Gupta et al., 2019). Best in **bold**, second-best underlined.

Anything-V2 (Yang et al., 2024) for metric mono-depth estimation. The dataset is filtered using methods from (Chen et al., 2024) to remove images on which the indoor-finetuned model performs poorly such as those with overlays, no visible groundplane, large outdoor scenes, or optical illusions.

We perform contrastive learning on the text to visual representation pairs in the WebLI-3D lifted dataset, where the visual representation may be in the form of an RGB image or an RGB-D depth image. Similar to CLIP (Radford et al., 2021), this is done by maximizing the similarity between the embeddings of matching visual-text pairs, while minimizing the similarity between embeddings of the non-matching pairs. This enables open-vocabulary detection and classification by comparing the text embeddings of all possible classes against those of the visual representation, and selecting the minimum distance pair. We compare against baseline in Table 3. **For all six evaluations**, Cayley-STRING is the best and Circulant-STRING is second best.

We demonstrate the efficacy of STRING on open-vocabulary object detection for localizing a 2D bounding box on standard RGB image benchmarks. For a baseline, we use the official implementation of OWL-ViT[4] (Matthias Minderer, 2022) which applies a standard Vision Transformer (Dosovitskiy et al., 2021) with light-weight classification and localization heads for detection. Table 4 compares the baseline OWL-ViT model with RoPE and STRING variants. For all experiments, we followed the standard training pipeline for the B/32 backbone with the CLIP loss (Radford et al., 2021). Even in this axis-aligned 2D detection setting – which is favourable for the standard RoPE variant – Cayley-STRING provides the best overall performance.

### 4.2.2. OPEN-VOCABULARY OBJECT DETECTION IN 3D

We tested STRING on the open-vocabulary 3D object bounding box prediction task, similar to those from Sec. 4.2.1. Here we modify the output to be the full SE(3) pose and 3D size of the bounding box. We initialize the weights of the vision and text towers of our model with the weights from the models trained on the WebLI-3D dataset from Sec. 4.1. We replace the IOU loss from OWL-ViT with an 8-corner vertex loss, but otherwise keep the same matching and loss algorithm. We train on a simulated dataset

---

[4] https://github.com/google-research/scenic/tree/main/scenic/projects/owl_vit

of 4 million images of indoor and tabletop scenes with groundtruth 3D bounding box labels (Lin et al., 2025) (see App.F.1 for details). From this dataset, we hold out 80 images for evaluation. We evaluate both ViT and ViTD variants. The 3D intersection-over-union (IOU) values for various RoPE and STRING variants on the evaluation data are shown in Table 5. For each configuration, we train 3 models with different random seeds and take the best performing model (see Appendix F.2.4 for details). Fig. 2 shows example 3D detections for 6 different variants (see App.F for details). Note that STRINGs provide much more accurate prediction of the 3D bounding boxes for more challenging to localize smaller objects than baseline and RoPE. For ViT, Circulant-STRING is the best, providing $1.5\%$ relative improvement as compared to the best RoPE variant. For ViTD, Cayley-STRING is the best, providing $2\%$ relative improvement as compared to the best RoPE variant. For both ViT and ViTD, two best models are STRINGs.

|  | Baseline | RoPE | RoPE-M | Circulant-S | Cayley-S |
|---|---|---|---|---|---|
| ViT | 49.77 | 58.09 | 57.17 | **58.95** | 58.85 |
| ViTD | 67.60 | 71.21 | 70.90 | 72.36 | **72.67** |

*Table 5.* Average 3D IOU % over all objects for the 3D bounding box prediction task. For each setting, 3 models were trained with different random seeds and the max is reported. Baseline indicates no RoPE or STRING. Best in **bold**, second-best underlined.

### 4.3. Simulation-Based Robot Manipulation: ALOHA

We evaluate the performance of STRING on dexterous robotics tasks using ALOHA 2, a bimanual parallel-jaw gripper workcell with two 6-DoF arms, within the ALOHA Unleashed (Zhao et al., 2024) simulation environment (see: Fig. 1). ALOHA Unleashed utilizes a scalable teleoperation framework used to collect human demonstration data.

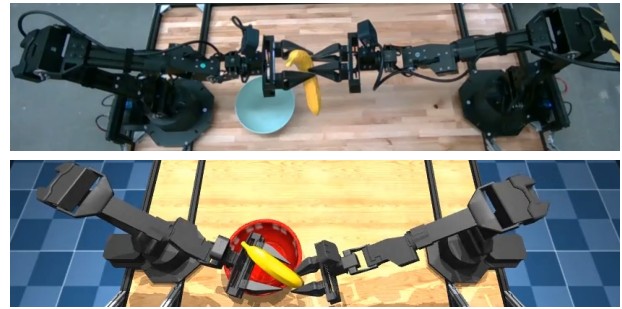

*Figure 3.* **HandOverBanana** task for the ALOHA 2 robot: real (top) and the corresponding simulated (bottom) evaluation.

We trained ALOHA Unleashed's Transformer-based neural network with diffusion policies (conditioned on vision encoders) on human demonstrations of 12 dexterous tasks (see Appendix B for descriptions and renders). The vision system utilized images from RGB cameras positioned overhead, on each wrist, and at the level of the table. We also deployed

| | Baseline | RoPE | Cayley-STRING |
|---|---|---|---|
| **ViT** | | | |
| **ViTD** | | | |

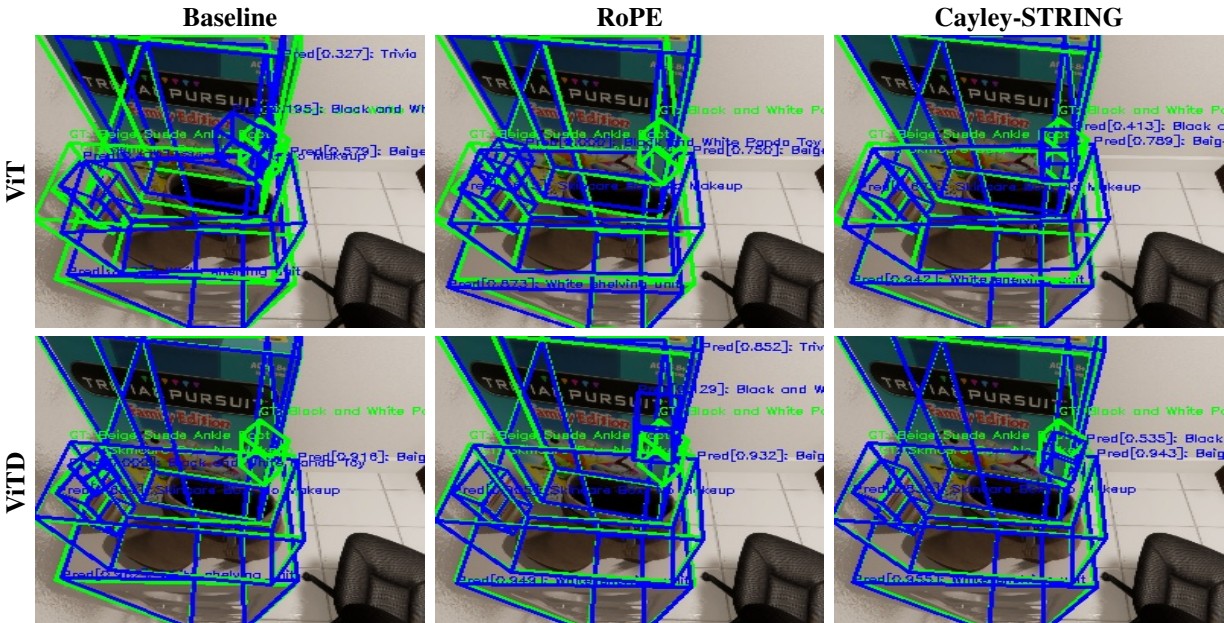

*Figure 2.* Example outputs for the 3D detection task for baseline, RoPE, and Cayley-S. Green boxes: groundtruth. Blue boxes: predictions.

our policies on real ALOHA 2 robots (see Fig. 3 and Fig. 11). Due to the large observed variance of the on-robot evaluation for ALOHA 2, we focused on the evaluation in simulation to accurately rank different methods.

Table 6 reports the best task success rate of using RoPE (Heo et al., 2025), and Cayley-STRING on a baseline SigLIP B/16 256 (Zhai et al., 2023) ViT model.

The success rate is averaged over 10 trials of each checkpoint, taken every 10K train steps and over 1M train steps. Corresponding curves are given in Fig. 4. Cayley-STRING achieves superior results across all tasks on average (i.e. MultiTask). Additionally, it achieves equivalent or superior results, as compared to RoPE (e.g. for the DoubleInsertion task from Fig. 1) and ViT for all 12 tasks except for MugOnPlate and PlateOnRack (second-best). Finally, STRING converges much faster than other methods (see: Fig. 4). Note that we applied the strategy of learning all angle-defining frequencies for both RoPE and Cayley-STRING.

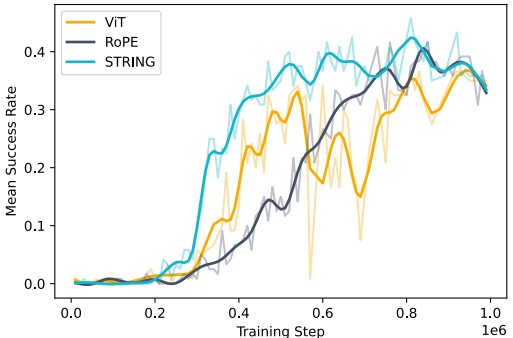

*Figure 4.* Mean success rate across all tasks (i.e. MultiTask) evaluated 10 times every 10K train steps over 1M train steps.

| | ViT | RoPE | STRING |
|---|---|---|---|
| BowlOnRack | 0.90 | 0.80 | **1.00** |
| DoubleInsertion | 0.20 | 0.50 | **0.60** |
| FMB-1 | **0.20** | **0.20** | **0.20** |
| FMB-2 | **0.10** | **0.10** | **0.10** |
| FruitBowl | **0.30** | **0.30** | **0.30** |
| GlassOnRack | **0.60** | **0.60** | **0.60** |
| HandOverBanana | **1.00** | **1.00** | **1.00** |
| HandOverPen | **1.00** | **1.00** | **1.00** |
| MugOnPlate | 0.70 | **0.90** | 0.80 |
| PlateOnRack | 0.60 | **0.70** | 0.50 |
| SingleInsertion | 0.40 | **0.60** | **0.60** |
| StorageBin | **0.00** | **0.00** | **0.00** |
| MultiTask | 0.37 | 0.42 | **0.46** |

*Table 6.* Mean success rate (best in **bold**, second-best underlined) over 10 evaluations of each ALOHA simulation task. RoPE (Heo et al., 2025) and Cayley-STRING are added to a baseline SigLIP B/16 256 ViT (Zhai et al., 2023). See Appendix B for details.

### 4.4. Real-World 3D Robot Manipulation: KUKA

Establishing STRING as superior to other methods on previous tasks, we let it operate on 3D data to obtain new SOTA robotic manipulation policies. This resulted in policies directly using depth and deployed on real hardware. Note that STRING can be naturally applied in that context since it can be used for data equipped with general coordinate vectors $\mathbf{r}_i$ associated with tokens (e.g. 3D).

#### 4.4.1. SETTING

We evaluated STRING in the vision encoder of a generative policy applying energy-based models (Singh et al., 2024) and deployed on a real industrial KUKA robot arm (Udayan

et al., 2023). The closed-loop feedback policy operates on RGB-D images, and is learned as a generative model with imitation learning. Its architecture (see Appendix G for details) consists of a diffusion Transformer and 3D encoders. The policy was trained on a mixture of scripted and teleoperated data collected for 3 different skills (pick, place and handover) on various objects. It is evaluated exclusively on the pick skill with a diverse set of objects. Each evaluation was run as an A/B test for a total of 50 trials.

### 4.4.2. REGULAR EVALUATIONS

We experimented with two ways of using depth in the policy.

**Implicit depth via normal maps:** In the first approach, following (Tziafas & Kasaei, 2023), depth input is used to construct a surface normal map with unit $\mathbb{R}^3$ values per pixel. Both RGB and depth inputs are then processed via identical (shared weights) embedding layers. The embeddings are concatenated and processed through Transformer layers. Finally, the embeddings are split and fused to yield the final vision embedding. Our results in Figure 5 show that this approach of incorporating depth has a detrimental effect on the on-robot deployed policy. We hypothesize that this is caused by the significant amount of noise coming from the depth sensors, leading to imperfect surface normal maps.

**Lifting patches to 3D for STRING:** In the second approach, we compute the height for each patch via mean-pooling across depth values for all the pixels in the patch, followed by the learnable linear layer. The resulting 3D patches are then fed to the Transformer, with positional encoding given by STRING to incorporate depth into the vision encoder. Our results Figure 5 show that STRING improves the success rate over the 2D base policy. Also, when STRING is combined with the first method, it **drastically** reduces the negative impact of noisy normal maps.

We used Circulant-STRING to obtain a particularly compact computational footprint. Note that in this setting, more computationally intense mechanisms, such as (Ostmeier et al., 2024), ran out of memory and could not be trained.

### 4.4.3. OUT-OF-DISTRIBUTION EVALUATION: STRING VS BASELINE

To further compare STRING with the baseline and show the advantages of using 3D over 2D policies, we also perform *out-of-distribution* (OOD) evaluations on the real robot.

We vary three different environment settings. These include: (1) lighting changes, (2) adding large distractor objects and (3) changing the height of the table from which the robot has to grasp the block. For each setting, we test multiple different variations, e.g., three different light settings.

Figure 6 compares STRING with the 2D baseline for each OOD setting. For these evaluations, we choose the best

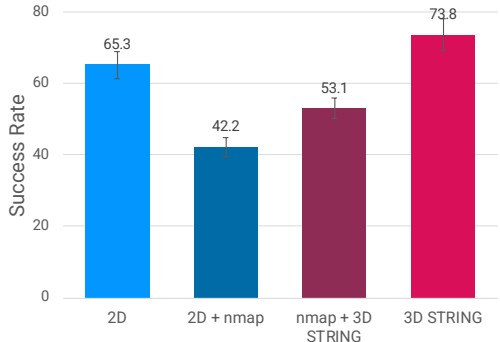

*Figure 5.* Performance of STRING with 3D input vs. baselines on real-robot tasks (with 2 seeds). 2D baseline performance without depth input is ≈ 65%. Incorporating depth through surface normal maps (nmap) reduces performance to 42%. Using 3D STRING for incorporating depth improves the performance in both scenarios - with and without normal maps to 53% and 74% respectively. Mean/stdev shown above were calculated from 35 evaluation runs.

policies from Section 4.4.2. As seen in Figure 6, 3D STRING policies outperform 2D policies across all OOD settings. For instance, with large distractors (middle), the 2D model's performance decreases from 65% to 57%, while 3D STRING *maintains* performance similar to non-OOD settings (≈ 74%). In some OOD cases, such as lighting changes, both 2D (≈ 10%) and 3D (≈ 25%) policies experience a performance decrease vs. the non-OOD setup. This drop in performance during lighting changes is likely due to the significant alteration in image observations, thus affecting both 2D and 3D policies. Finally, the largest performance difference is observed in the table height variation scenario. Here, the 3D policies exhibit significantly higher robustness (≈ **50%**) compared to the 2D policies (≈ **10%**). This suggests that the 3D STRING policy leverages the raw depth signal to better generalize to table height variations, a change imperceptible to fixed monocular cameras.

Overall, our results show that 3D STRING policies are highly robust to many variations and significantly improve over 2D policies. Fig. 7 shows a sample episode from the on-robot evaluation of the STRING generative policy.

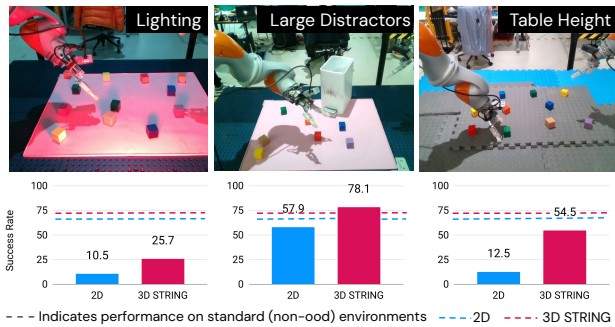

*Figure 6.* Comparison of 2D baseline with 3D STRING on out-of-distribution scenarios for real-world Kuka robot evaluations.

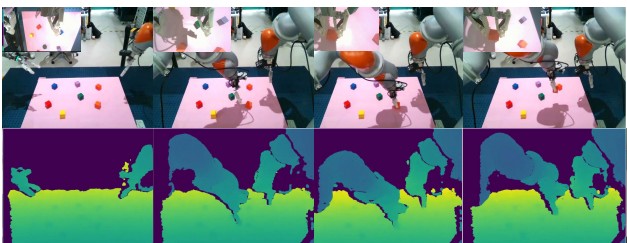

*Figure 7.* Rollout frames from a successful policy evaluation in the real world (top: RGB, bottom: depth). Best viewed zoomed-in.

**From 2D to 3D with STRING:** We have already demonstrated (see the normal map approach from Section 4.4.2) that just adding a depth signal does not necessarily improve performance. STRING does so and exhibits another feature that other approaches (e.g. adding depth as extra channel) do not: it can be trained from a regular 2D pre-trained checkpoint. This is the case since STRING incorporates depth by using it to modulate a regular attention matrix, effectively disentangling 3D specific parameters (defining the modulation) from the core 2D backbone. All training runs in Section 4.4 started from the pre-trained 2D backbones.

## 5. Conclusion

We introduced a new class of translationally invariant position encodings (PEs) for Transformers, called STRING. STRING is the most general of all translation-invariant PE methods using matrix multiplications (under weak smoothness assumptions) and contains the prominent class of RoPE methods as its special instantiation. We proposed to apply STRING in robotics for 2D and 3D modeling and provided its extensive empirical verification over a range of tasks, from standard classification and retrieval, through object localization, to diffusion robotic policies conditioned on Vision Transformers. In all these experiments, we showed consistent gains over RoPE, as well as baselines applying regular absolute position encodings.

## Impact Statement

The goal of this work is to contribute to the advancement of the machine learning field which may result in potential societal consequences. We acknowledge these potential risks, especially in downstream use-cases of advanced machine learning techniques and advocate for careful consideration of ethical implications in the development and deployment of these techniques. Additionally, as it is the case for all papers discussing training Transformer architectures, the corresponding carbon footprint needs to be taken into account. STRING plays a positive role here since it reduces computational costs by providing ways of fine-tuning already pre-trained architectures with a negligible set of extra trainable parameters.

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

# A. Proofs

## A.1. Proof of Theorem 3.2

To begin, we provide a proof of Theorem 3.2: that STRING is the most general form of transformation that respects the group-like property

$$\mathbf{R}(\boldsymbol{r}_i)^\top \mathbf{R}(\boldsymbol{r}_j) = \mathbf{R}(\boldsymbol{r}_j - \boldsymbol{r}_i), \tag{15}$$

supposing that $\mathbf{R}(\mathbf{0}) = \mathbf{I}_d$ and $\mathbf{R}(\mathbf{r})$ is continuously differentiable with respect to $\mathbf{r}$. Note that this is a sufficient, but not necessary, assumption for translational invariance.

*Proof.* Recall STRING is applied as

$$\mathbf{q}_i \to \mathbf{R}(\boldsymbol{r}_i)\mathbf{q}_i \tag{16}$$

with $\mathbf{R}(\cdot) : \mathbb{R}^{d_c} \to \mathbb{R}^{d \times d}$, so that $\mathbf{R}(\boldsymbol{r}_i) \in \mathbb{R}^{d \times d}$. Then we require that

$$\mathbf{q}_i^\top \mathbf{k}_j \to \mathbf{q}_i^\top \mathbf{R}(\boldsymbol{r}_i)^\top \mathbf{R}(\boldsymbol{r}_j)\mathbf{k}_j. \tag{17}$$

Recall that $\mathbf{R}(\mathbf{0}) = \mathbf{I}_d$, the $d$-dimensional identity, so that the logit of a query and key at the same position ($\boldsymbol{r}_i = \boldsymbol{r}_j$) will be unmodified by the positional encoding. Then clearly $\mathbf{R}(\boldsymbol{r}_i) \in \mathrm{O}(d)$, the orthogonal group in dimension $d$. For compatability with gradient-based optimisers (a chief concern in the machine learning setting), it is convenient to specialise to the connected component (normal subgroup) of $\mathrm{O}(d)$ containing the identity matrix: that is, the special orthogonal group $\mathrm{SO}(d)$.[5] This means that $\det(\mathbf{R}(\boldsymbol{r}_i)) = 1$. These transformations are the $d$-dimensional rotations.

Since $\mathbf{R}(\boldsymbol{r}_i) \in \mathrm{SO}(d)$, the rotation can be written using its Lie group,

$$\mathbf{R}(\boldsymbol{r}_i) = \exp(\mathbf{L}(\boldsymbol{r}_i)) \tag{18}$$

where the matrix $\mathbf{L}(\boldsymbol{r}_i)$ is antisymmetric (Hall, 2013). $\mathbf{L}(\boldsymbol{r}_i)$ is called the 'generator', representing an infinitesimal rotation. Here, $\exp(\cdot)$ denotes the *matrix* exponential (not to be confused with the element-wise exponential of a matrix, as appears e.g. in softmax). Setting $\boldsymbol{r}_j = \mathbf{0}$ in Equation (15), it is clear that $\mathbf{L}(-\boldsymbol{r}_i) = -\mathbf{L}(\boldsymbol{r}_i)$. We then require that

$$\exp(\mathbf{L}(\boldsymbol{r}_i))\exp(\mathbf{L}(\boldsymbol{r}_j)) = \exp(\mathbf{L}(\boldsymbol{r}_i + \boldsymbol{r}_j)) \\ = \exp(\mathbf{L}(\boldsymbol{r}_j))\exp(\mathbf{L}(\boldsymbol{r}_i)). \tag{19}$$

Clearly $\mathbf{L}(\boldsymbol{r}_i)$ and $\mathbf{L}(\boldsymbol{r}_j)$ must commute for all choices of coordinate vector $(\boldsymbol{r}_i, \boldsymbol{r}_j)$. Therefore, we need

$$\mathbf{L}(\boldsymbol{r}_i + \boldsymbol{r}_j) = \mathbf{L}(\boldsymbol{r}_i) + \mathbf{L}(\boldsymbol{r}_j), \tag{20}$$

so $\mathbf{L}(\cdot)$ is linear in its arguments. That is, $\mathbf{L}(\cdot)$ is a linear map from the set of $d_c$-dimensional vectors to a set of

---

[5] This means that you can optimise the position encoding transformations on the same manifold. You could in prinicple also incorporate reflections so that $\det(\mathbf{R}(\boldsymbol{r}_i)) = -1$, but this seems unlikely to significantly improve performance.

commuting antisymmetric matrices. We can write

$$\mathbf{L}(\boldsymbol{r}_i) = \sum_{k=1}^{d_c} \mathbf{L}_k[\boldsymbol{r}_i]_k, \tag{21}$$

with $\{\mathbf{L}_k\}_{k=1}^{d_c} \subset \mathbb{R}^{d \times d}$ a set of *commuting antisymmetric generators* and $[\boldsymbol{r}_i]_k$ the $k$-th entry of coordinate vector $\boldsymbol{r}_i$. This completes the proof. □

## A.2. Proof of Theorem 3.3

Now, we prove that generators of the form $\mathbf{L}_k = \sum_{p=1}^{d/2}(\delta_{2p,2p-1} - \delta_{2p-1,2p})\theta_p$ recover RoPE, as described in Theorem 3.3.

*Proof.* Let us initially consider the case $d_c = 1$, so that the token coordinate $\mathbf{r} = r \in \mathbb{R}$ and we learn a single generator. Recall that powers of a block diagonal matrix will remain block diagonal. Each block of the generator $\mathbf{L}_k$ is of the form

$$\begin{bmatrix} 0 & \theta \\ -\theta & 0 \end{bmatrix}. \tag{22}$$

Then note that

$$\begin{bmatrix} 0 & \theta \\ -\theta & 0 \end{bmatrix}^2 = \begin{bmatrix} -\theta^2 & 0 \\ 0 & -\theta^2 \end{bmatrix}. \tag{23}$$

It follows that

$$\begin{bmatrix} 0 & \theta \\ -\theta & 0 \end{bmatrix}^n = \begin{cases} \theta^n (-1)^{n/2} \begin{bmatrix} 1 & 0 \\ 0 & 1 \end{bmatrix} & \text{if } n \text{ is even,} \\ \theta^n (-1)^{(n-1)/2} \begin{bmatrix} 0 & 1 \\ -1 & 0 \end{bmatrix} & \text{if } n \text{ is odd.} \end{cases} \tag{24}$$

Combining and inspecting the Taylor expansions,

$$\exp \begin{bmatrix} 0 & \theta \\ -\theta & 0 \end{bmatrix} = \begin{bmatrix} \cos\theta & \sin\theta \\ -\sin\theta & \cos\theta \end{bmatrix}, \tag{25}$$

which is clearly a rotation matrix – a well-known result. This holds for all the $d/2$ blocks, each of which exponentiates to give a $2 \times 2$ rotation at a different frequency. Therefore,

$$\exp(\mathbf{L}r) = \mathrm{RoPE}(r), \tag{26}$$

showing that with this particular generator STRING is RoPE.

Now suppose that $d_c > 1$, so $\mathbf{L}(\boldsymbol{r}_i) = \sum_{k=1}^{d_c} \mathbf{L}_k[\boldsymbol{r}_i]_k$. In our special case, each generator is of the form $\mathbf{L}_k = \sum_{p=1}^{d/2}(\delta_{2p,2p-1} - \delta_{2p-1,2p})\theta_p$, where $\{\theta_p\}_{p=1}^{d/2}$ can differ for different $k$ (notationally suppressed for compactness). Observing that

$$\begin{bmatrix} 0 & \alpha \\ -\alpha & 0 \end{bmatrix}\begin{bmatrix} 0 & \beta \\ -\beta & 0 \end{bmatrix} = \begin{bmatrix} -\alpha\beta & 0 \\ 0 & -\alpha\beta \end{bmatrix} \\ = \begin{bmatrix} 0 & \beta \\ -\beta & 0 \end{bmatrix}\begin{bmatrix} 0 & \alpha \\ -\alpha & 0 \end{bmatrix}, \tag{27}$$

different $\mathbf{L}_k[\mathbf{r}_i]_k$ commute. Then we have that

$$\exp\left(\sum_{k=1}^{d_c}\mathbf{L}_k[\mathbf{r}_i]_k\right) = \prod_{k=1}^{d_c}\exp\left(\mathbf{L}_k[\mathbf{r}_i]_k\right)$$
$$= \prod_{k=1}^{d_c}\text{RoPE}([\mathbf{r}_i]_k) \qquad (28)$$
$$= \text{RoPE}(\mathbf{r}),$$

where we used the definition for multidimensional RoPE from Equation (5). This completes the proof. $\qquad\square$

### A.3. Proof of Theorem 3.4

Next, we prove that STRING can always be rewritten as RoPE in a different basis.

*Proof.* As usual, we begin with $d_c = 1$. Then our task is to show that a matrix $\mathbf{R}(r) = \exp(\mathbf{L}r)$, with $\mathbf{L} \in \mathbb{R}^{d\times d}$ an antisymmetric matrix and $r \in \mathbb{R}$, can be rewritten as RoPE with a change of basis.

Begin by noting that $\mathbf{R}(r) \in SO(d)$; it is a special orthogonal matrix. It is orthogonal since $\mathbf{R}(r)^\top\mathbf{R}(r) = \exp(\mathbf{L}r)^\top\exp(\mathbf{L}r) = \exp(-\mathbf{L}r)\exp(\mathbf{L}r) = \mathbf{I}_d$, and its determinant is 1 since it is continuously connected to the identity (which occurs at $r = 0$). Consider an eigenvector $\mathbf{v} \in \mathbb{C}^d$, with eigenvalue $\lambda \in \mathbb{C}$. Since $\mathbf{R}^\top\mathbf{R} = \mathbf{I}_d$, $|\lambda| = 1$ so $\lambda = e^{i\theta}$. Taking the complex conjugate of $\mathbf{R}\mathbf{v} = \lambda\mathbf{v}$, we have $\mathbf{R}\bar{\mathbf{v}} = \lambda^*\bar{\mathbf{v}}$, where $\lambda^* = e^{-i\theta}$ and $\bar{\mathbf{v}}$ is the complex conjugate of $\mathbf{v}$. We used that $\mathbf{R}$ is real. Therefore, the eigenvalues appear in conjugate pairs for conjugate eigenvectors.

Let $\mathbf{v} = \mathbf{u} + i\mathbf{w}$ and $\bar{\mathbf{v}} = \mathbf{u} - i\mathbf{w}$ with $\mathbf{u}, \mathbf{w} \in \mathbb{R}^d$ real vectors. Inserting into the eigenvector equation, $\mathbf{R}(\mathbf{u} + i\mathbf{w}) = (\cos(\theta) + i\sin(\theta))(\mathbf{u} + i\mathbf{w})$. Equating the real and imaginary parts, $\mathbf{R}\mathbf{u} = \cos(\theta)\mathbf{u} - \sin(\theta)\mathbf{w}$ and $\mathbf{R}\mathbf{v} = \cos(\theta)\mathbf{v} + \sin(\theta)\mathbf{u}$. This corresponds exactly to 2-dimensional rotation in the $(\mathbf{u}, \mathbf{v})$ plane. By normalisation of (complex) $\mathbf{v}$, $|\mathbf{u}|^2 + |\mathbf{w}|^2 = 1$. But since $\lambda$ and $\lambda^*$ differ, their corresponding eigenvectors are orthogonal under the Hermitian inner product, so we also have that $\bar{\mathbf{v}}^\dagger\mathbf{v} = |\mathbf{u}|^2 - |\mathbf{w}|^2 + 2i\mathbf{u}^\top\mathbf{w} = 0$. So $\mathbf{u}^\top\mathbf{w} = 0$, whereupon $\mathbf{u}$ and $\mathbf{w}$ are orthogonal vectors in $\mathbb{R}^d$. Of course, from basic linear algebra the complex eigenvectors corresponding to *different* $\theta$ will also be orthogonal in $\mathbb{C}^d$, or in the case of degenerate $\theta$ one can choose an orthogonal basis for the corresponding subspace using e.g. the Gram-Schmidt process. It is easy to show that the corresponding real vectors $(\mathbf{u}_i, \mathbf{w}_i)$ will therefore be orthogonal for different $i$, i.e. $\mathbf{u}_i^\top\mathbf{u}_j = \delta_{i,j} = \mathbf{w}_i^\top\mathbf{w}_j$. To summarise: the real and imaginary parts of the the $d$ orthogonal (complex) eigenvectors of $\mathbf{R}$ in $\mathbb{C}^d$ correspond to $d$ orthogonal (real) vectors in $\mathbb{R}^d$, organised in $\frac{d}{2}$ planes in each of which $\mathbf{R}$ acts as a 2-dimensional rotation, at a frequency that depends on the corresponding eigenvalue. These real vectors $\{\mathbf{u}_i, \mathbf{w}_i\}_{i=1}^{d/2}$

can be aggregated as the columns an orthogonal matrix $\mathbf{P}$, taking

$$\mathbf{P} := \begin{bmatrix} \uparrow & \uparrow & \uparrow & \uparrow & \cdots \\ \mathbf{u}_1 & \mathbf{w}_1 & \mathbf{u}_2 & \mathbf{w}_2 & \cdots \\ \downarrow & \downarrow & \downarrow & \downarrow & \cdots \end{bmatrix} \in \mathbb{R}^{d\times d} \qquad (29)$$

whereupon

$$\mathbf{R}(r) = \mathbf{P}\text{RoPE}(r)\mathbf{P}^\top. \qquad (30)$$

Note especially that the change of basis matrix $\mathbf{P}$ is independent of $r$, because $\mathbf{P}$ determines the axes of rotation whereas the (complex) eigenvalues depend on the amount of rotation. This is obvious from the definition $\mathbf{R}(r) = \exp(\mathbf{L}r)$; the matrix $\mathbf{P}$ needs to (block) diagonalise $\mathbf{L}$, then this is sufficient to (block) diagonalise $\mathbf{R}$, and the matrix $\mathbf{L}$ is independent of $\mathbf{r}$.

Now suppose that $d_c > 1$. Recall that we have

$$\mathbf{R}(\mathbf{r}) = \exp\sum_{k=1}^{d_c}\mathbf{L}_k\mathbf{r}_k = \prod_{k=1}^{d_c}\exp\mathbf{L}_k\mathbf{r}_k = \prod_{k=1}^{d_c}\mathbf{R}([\mathbf{r}]_k),$$
$$(31)$$

where we used the fact that the generators $\{\mathbf{L}_k\}_{k=1}^{d_c}$ commute (from Definition 3.1). $\{\mathbf{R}(\mathbf{r}_k)\}_{k=1}^{d_c}$ must also commute so are simultaneously diagonalisable, whereupon

$$\mathbf{R}(\mathbf{r}) = \prod_{k=1}^{d_c}\mathbf{P}\text{RoPE}([\mathbf{r}]_k)\mathbf{P}^\top$$
$$= \mathbf{P}\left(\prod_{k=1}^{d_c}\text{RoPE}([\mathbf{r}]_k)\right)\mathbf{P}^\top \qquad (32)$$
$$= \mathbf{P}\text{RoPE}(\mathbf{r})\mathbf{P}^\top.$$

This concludes the proof. $\qquad\square$

### A.4. Proof of Theorem 3.5

*Proof.* Note that it suffices to show that the computation of $\mathbf{p} = \exp(\mathbf{C} - \mathbf{C}^\top)\mathbf{z}$ can be computed in $\mathcal{O}(d\log(d))$ time with $\mathcal{O}(d)$ memory for a circulant matrix $\mathbf{C}$, defined by its first row $\mathbf{c}$. We will leverage the fact that every circulant matrix $\mathbf{C}$ can be factorized as follows:

$$\mathbf{C} = \mathbf{DFT} \times \text{diag}(\mathbf{DFT}\mathbf{c}) \times \mathbf{DFT}^{-1}, \qquad (33)$$

for the Discrete Fourier Transform matrix $\mathbf{DFT}$, and where $\text{diag}(\mathbf{v})$ stands for the diagonal matrix with the main diagonal given by $\mathbf{v}$. Therefore we have the following:

$$\mathbf{p} = \exp(\mathbf{DFT}(\text{diag}(\mathbf{DFT}\mathbf{u}))\mathbf{DFT}^{-1})\mathbf{z}, \qquad (34)$$

where $\mathbf{u} = \mathbf{c} - \mathbf{t}$ and $\mathbf{t}$ stands for the first column of $\mathbf{C}$. Here we leverage the fact that the transpose of the circulant matrix is also circulant. Therefore, by leveraging Taylor

series formula for exp, we get:

$$\mathbf{p} = \mathbf{DFT}\exp(\text{diag}(\mathbf{DFTu}))\mathbf{DFT}^{-1}\mathbf{z}$$
$$= \mathbf{DFT}\text{diag}(\exp(\mathbf{DFTu}))\mathbf{DFT}^{-1}\mathbf{z}, \quad (35)$$

where exp in the last formula is computed element-wise. Note that matrix-vector product with diagonal matrices can be trivially conducted in linear time. Thus the calculation of $\mathbf{p}$ can be conducted in $\mathcal{O}(d)$ memory and $\mathcal{O}(d\log(d))$ time complexity via Fast Fourier Transform (FFT) (to multiply with $\mathbf{DFT}$ matrices) and inverse FFT (iFFT) (to multiply with matrices $\mathbf{DFT}^{-1}$). This completes the proof. $\square$

## B. ALOHA Unleashed Simulation Tasks

ALOHA leader robots are teleoperated in ALOHA simulation for data collection using an ALOHA station, an Oculus VR headset and controllers (Zhao et al., 2024). The teleoperators are instructed to perform the following tasks using this setup. See Figure 9 and Figure 10 for simulation renders of the following 12 tasks.

1-3. `{Bowl/Glass/Plate}OnRack`: Place the item on the rack.

4. `SingleInsertion`: Use the left arm to grab the blue socket and the right arm to insert the block into the socket.

5. `DoubleInsertion`: After completing the `SingleInsertion` task, insert another block into the other end of the socket.

6-7. Functional Manipulation Benchmark (`FMB`) 1 and 2: Insert a yellow block into the recess of a red base.

8. `FruitBowl`: Place all the fruits in the bowl.

9. `StorageBin`: Place all the snack boxes in the storage bin.

10-11. `HandOver{Banana/Pen}`: Hand over the item and place it in the container.

12. `MugOnPlate` Place the mug on the plate.

`MultiTask` aggregates results of all of the above tasks.

## C. ALOHA Real Tasks

ALOHA-real models are first pre-trained with human-teleop data collected on 300 diverse tasks, which were crowdsourced based on relevance with real-world scenarios as well as feasibility for the ALOHA robot. Further, the model is fine-tuned on the following 10 tasks.

1. `open-jar-lid`: open the glass jar lid, handover to other hand and put on the table

2. `bowl-in-rack`: put the bowl into the drying rack

3. `cup-in-rack`: put the cup into the drying rack

4. `banana-handover`: put banana in bowl with handover

5. `open-drawer`: open the drawer

6. `remove-gears`: remove the gears from the nistboard

7. `fold-dress`: fold the dress

8. `stack-cups`: stack the cups

9. `pen-handover`: put pen in container with handover

10. `take-phone-out`: take phone out of purse

After fine-tuning, the model is then evaluated on the following 5 tasks: `bowl-in-rack`, `banana-handover`, `bowl-in-rack`, `fold-dress`, `remove-gears`. See: Fig. 11 for the visualizations of selected ALOHA real world tasks.

## D. WebLI-3D Dataset

We lift the WebLI dataset (Chen et al., 2023), a dataset of 10 billion image-text pairs across a variety of languages, into 3D by pre-processing a 60-million image subset using Depth-Anything-V2 (Yang et al., 2024) for metric monodepth estimation. The dataset is filtered using the method described in (Chen et al., 2024) for images that the indoor-finetuned model performs poorly on (pictures with overlays, no visible groundplane, large outdoor scenes, optical illusions are removed).

## E. Classification and Retrieval Experiment Details

Our experimental base model is ViT-B/16 (Dosovitskiy et al., 2021), which has 12 layers, 768 model dimension, 3072 MLP size, 12 attention heads, and $16 \times 16$ patch size. All RoPE and STRING variants retain these same model hyperparameters. For vanilla RoPE, we use the common 10,000 max wavelength and simply split query/key dimensions between 2 or 3 axes in the 2D or 3D case, respectively. For RoPE-Mixed, we initialize with 100 max wavelength as suggested in Heo et al. (2025), which we also adopt for STRING.

For other hyperparameters like learning rate, batch size, warm-up schedule, etc., we left these the same as the default values used for the base ViT-B/16 model. We use

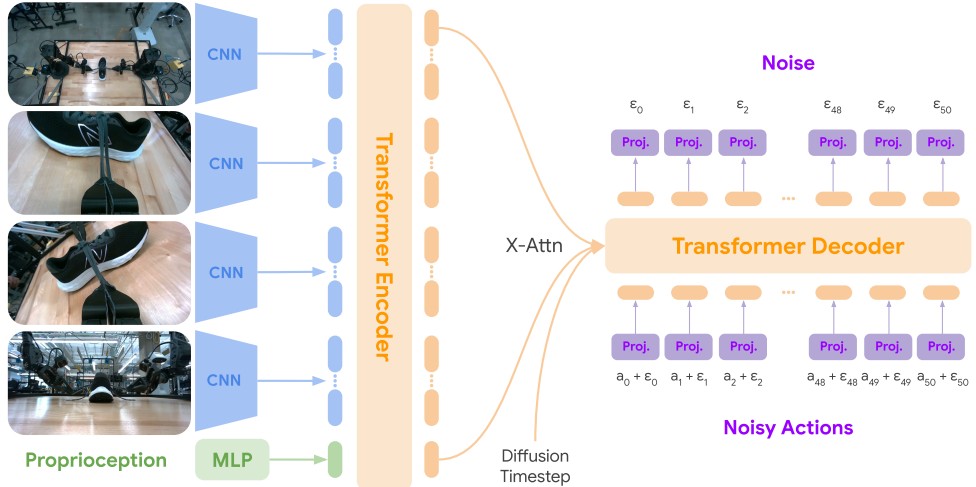

*Figure 8.* ALOHA Unleashed Architecture (Zhao et al., 2024).

the Adam optimizer with b1=0.9 and b2=0.999 for all experiments. For STRING, we experimented with sharing parameters across attention heads rather than the default of learning them all separately, and we found this could yield slight gains in both efficiency and quality. Finally, for Circulant-STRING, we swept over block sizes in the set $\{4, 8, 16, 32, 64\}$ to find the optimal setting (often around 16).

### E.1. ImageNet2012 and Places365 Classification

All experiments were trained from scratch, separately for ImageNet2012 or Places365. In both cases, we used 224×224 image resolution, batch size 4096, and trained for 300 epochs. Training used the cosine decay learning rate schedule with 0.001 base learning rate and 10,000 warm-up steps. For ImageNet2012 there were a total of about 94k training steps, and for Places365 there were about 130k. For Circulant-STRING, block size 16 yielded the best results.

### E.2. WebLI-3D Retrieval

All experiments were trained from scratch and used 256×256 image resolution. The ViT baseline used RGB data, but all RoPE and STRING variants used RGB-D, where we incorporated depth as a third coordinate for 3D position representations. We trained with batch size 8192 for 20 epochs, amounting to about 155k training steps using the SigLIP (Zhai et al., 2023) pretraining setup. Training used the cosine decay learning rate schedule with 0.001 base learning rate and 5% warm-up steps (about 8k). For Circulant-STRING, block size 32 yielded the best results.

## F. 3D Detection Details

### F.1. Dataset

We train our synthetic datasets of indoor living room and cluttered tabletop scenes using a procedural generation recipe. We utilize the dataset described in (Lin et al., 2025), which we briefly describe here. We use open-sourced 3D assets, specifically a subset of assets from the Amazon Berkeley Objects (Collins et al., 2022) (ABO) dataset for background and tabletop objects and the YCB (Calli et al., 2015) and Google Scanned Objects (Downs et al., 2022) for tabletop clutter placement. The procedural generation recipe works by hierarchically generating sub-areas such as a lounge, dining, office and reading area within a randomly-sized rectangular room together with freestanding pieces of furniture, sampled from the different classes within the ABO assets. For scenes with tabletop clutter, we additionally randomly sample assets on top of existing placement surfaces (e.g. tables) in the scene, and procedurally vary the packing fraction on top of the placement surface to achieve randomised clutter arrangements. We additionally vary lighting, background colors, camera extrinsics, intrinsics, aspect ratios etc to generate diverse datasets. Each example within a dataset has accompanied metadata such as RGB, Depth, Segmentation masks, Object poses, Camera extrinsics, intrinsics, and 3D & 2D bounding boxes for each object in view. We render images using Unity (Unity Technologies, 2023) achieving a high-degree of photo-realism.

We generated four different 1-million image datasets using the procedure above: a) living room scenes without table-top clutter (Fig. 12a), b) tabletop scenes with procedurally varying clutter, placed within randomly created living room scenes (Fig. 12b), c) tabletop scenes with multiple cluttered tabletops (Fig. 12c), and d) tabletop scenes with more com-

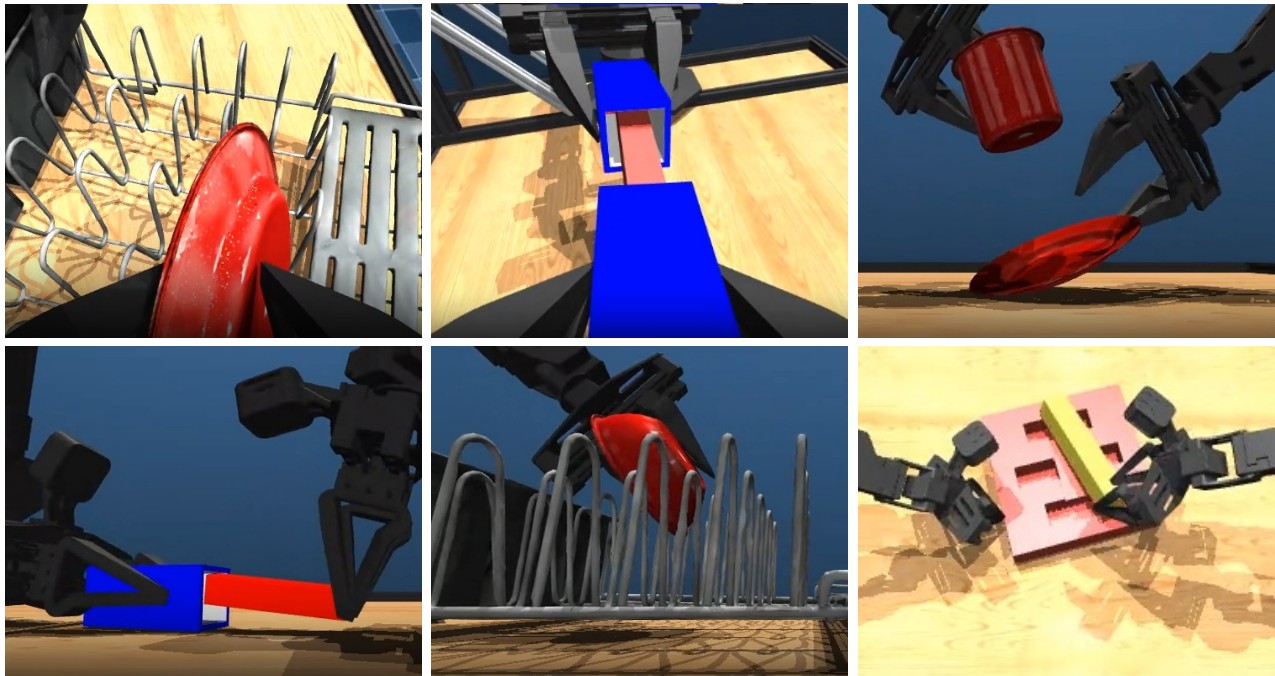

*Figure 9.* Views from the wrist, overhead and table-level cameras in ALOHA sim. **Top to bottom, Left to right:** PlateOnRack, DoubleInsertion (insert peg into sockets on either end), MugOnPlate, SingleInsertion, BowlOnRack, and Functional Manipulation Benchmark-1 (FMB-1).

plicated object arrangements, primarily objects placed on top of trays and other flat objects within the scene (Fig. 12d). We held out the first 20 images for each of these datasets for evaluation, and used the rest for training. Example images from our datasets showing the images and corresponding 3D bounding box labels can be seen in Fig. 13.

### F.2. Implementation

We base our implementation for 3D bounding box detection described in Section 4.2.2 on OWL-ViT (Matthias Minderer, 2022). OWL-ViT predicts 2D, axis-aligned bounding boxes for the queried object names in the image. We modify this to predict full 3D bounding boxes. We utilized the vision and text towers from the SigLIP task in Section 4.1, and add the box and class prediction heads on top for the 3D bounding box task. The main differences with the original OWL-ViT are described in the sections below.

#### F.2.1. BOX FORMAT

The output of the box head in OWL-ViT is the relative offset from the corresponding image patch for the 4 edges (top, bottom, left, and right) of the axis-aligned, 2D bounding box. In 3D, image patches do not align with predictions quite as well as in 2D, so instead, the box head predicts the absolute pose of the 3D bounding box. The output format

of the box head is

$$[< translation >, < rotation >, < size >]$$

where $< translation > \in \mathbb{R}^3$ is the SE(3) translation of the center of the box, $< rotation > \in \mathbb{R}^6$ is the first 2 columns of the SO(3) rotation matrix of the box, and $< size > \in \mathbb{R}^3$ is the length of each side of the box. Thus the output of the model for each predicted box is a 12D vector. All predictions are relative to the camera frame.

#### F.2.2. LOSS FUNCTION

We modify the OWL-ViT loss terms in the following way. We keep the class loss as-is. We also keep the L1 loss directly on the 12D bounding box prediction vector. However, this loss is not necessarily optimal for SO(3) rotations and in future work we would like to look into better rotation losses such as (Hertzberg et al., 2011). Finally, we completely replace the 2D, axis-aligned intersection-over-union (IOU) loss. The algorithm for computing the full 3D, non-axis-aligned IOU is non-differentiable, so we instead compute a loss over the 8 corner vertices of the box. For both the predicted and target boxes, we compute the 3D coordinates of the 8 corners of the boxes, and then we sum the L1 distance between the predicted and target corners (we assume a fixed ordering of the corners). We found this loss significantly increases overall performance of the learned models.

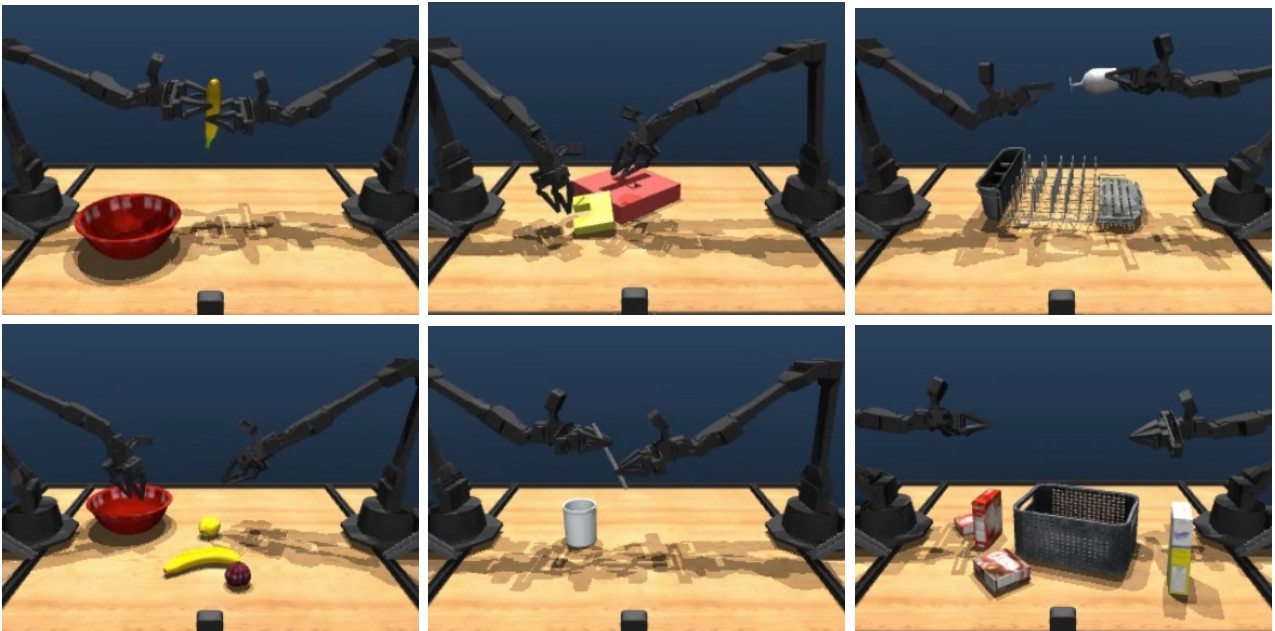

*Figure 10.* More ALOHA Simulation Tasks. **Top to bottom, left to right:** FruitBowl, FMB-2, GlassOnRack, HandOverBanana, HandOverPen and StorageBin.

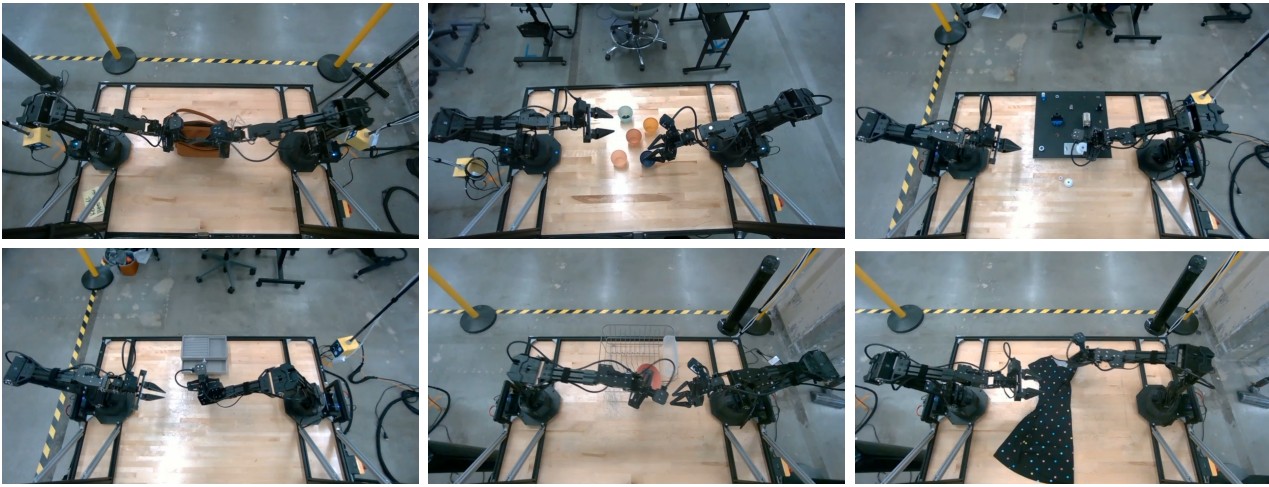

*Figure 11.* Sampled ALOHA Real Tasks. **Top to bottom, left to right:** take-phone-out, stack-cup, remove-gear, open-drawer, bowl-in-rack and fold-dress.

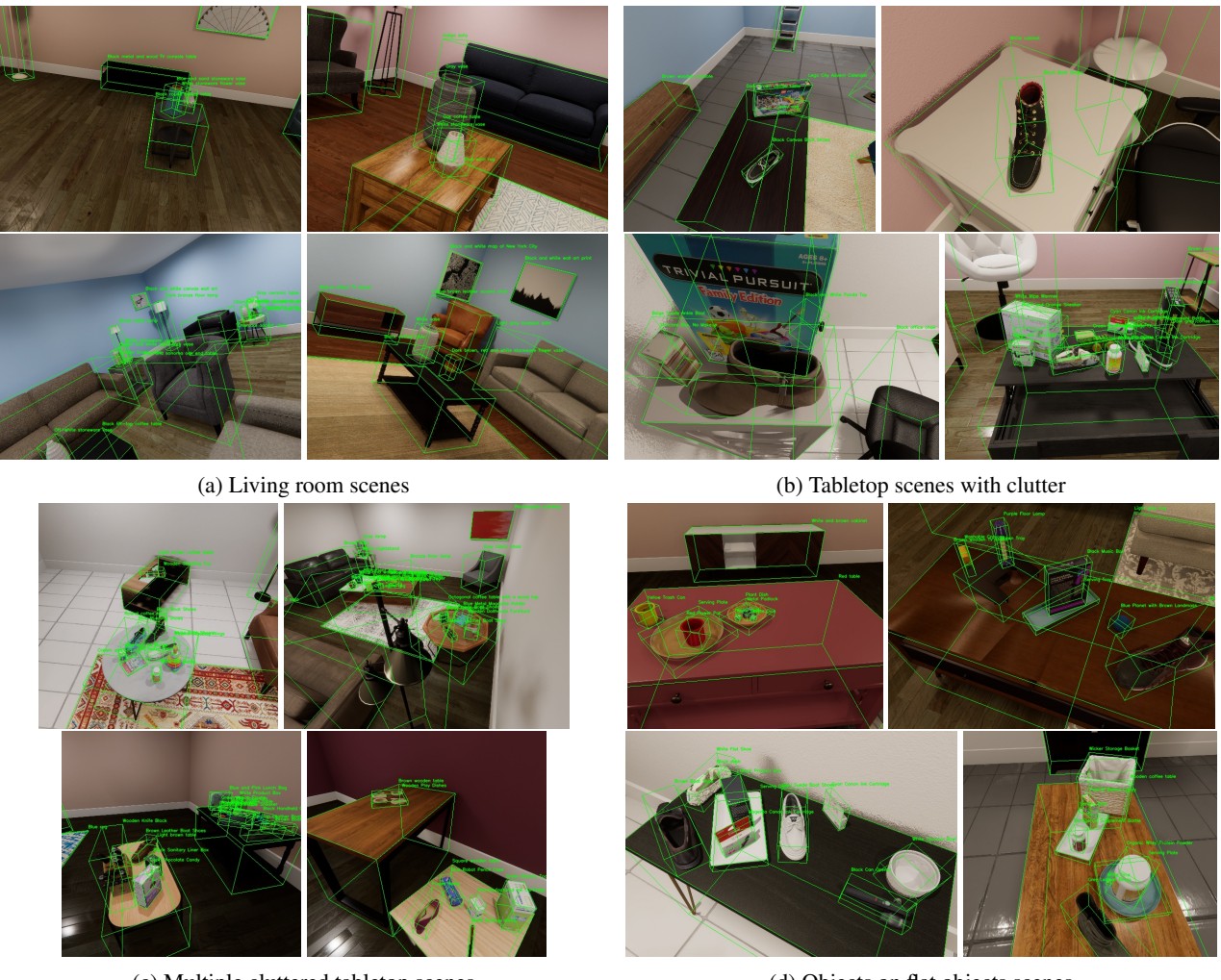

(a) Living room scenes

(b) Tabletop scenes with clutter

(c) Multiple cluttered tabletop scenes

(d) Objects on flat objects scenes

*Figure 13.* Example images drawn at random from our dataset used for 3D detection. Each subfigure shows 4 images from each of the 4 generated subsets of the dataset. The green boxes show the ground truth object bounding boxes.

### F.2.3. 3D IOU

We use a Monte Carlo algorithm to compute the intersection-over-union (IOU) between 2 boxes in 3D space, which we report in our evaluation numbers in Section 4.2.2. First, we sample 100k 3D points uniformly at random inside the predicted box. Next, we transform those points to be in the coordinate space of the target box and we compute the ratio of points that are inside the target box. Finally, we compute IOU as the intersection of the volumes divided by the sum of the volumes minus the intersection:

$$IOU(box_p, box_t) = \frac{r * vol(box_p)}{vol(box_t) + (1 - r) * vol(box_p)}$$

where $r$ is the fraction of sampled points inside the target box and $vol$ is the volume of the given box.

### F.2.4. BIPARTITE MATCHER

We found empirically that the Hungarian algorithm (Kuhn, 1955) for bipartite matching between predictions and targets during training had the best performance. Specifically, the Hungarian algorithm was the only bipartite matching algorithm we tested that was able to correctly detect objects far from the center of the image. We suspect that, due to the absolute bounding box predictions described above in Appendix F.2.1, the box head was biased towards objects in the center of the camera frame, and the Hungarian algorithm encouraged it to predict boxes for further away objects. However, we also found the Hungarian algorithm to produce significant volatility in the performance of the model, with models sometimes getting stuck in local minima with poor performance during training (see Table 7). To counteract this, for every entry in Table 5 in Section 4.2.2, we trained 3 models with different random seeds. The values in the table are the max of the 3 models' performance after 250k training iterations. Note that the parameters for the vision and text towers are loaded from the pre-trained checkpoints, so only the parameters for the prediction heads are randomly initialized.

### F.3. Model and Training Configuration

The model is composed of 4 parts: the vision tower, the text tower, the class head, and the box head. The vision tower encodes the image as a set of tokens, the text tower encodes each input text sequence as a token, the class head predicts the class probabilities for the predictions given the query texts, and the box head outputs the bounding box parameters for each prediction. For the vision and text towers, we use the same model layout (e.g., number of layers) as the models trained on WebLI-3D in Section 4.1. For the class head we use the same layout as described in OWL-ViT (Matthias Minderer, 2022). We modify the box head to be a 3 layer MLP, with GELU (Hendrycks & Gimpel, 2016) non-linearities after the first 2 layers. Unlike OWL-ViT, which

outputs a relative offset for the bounding box, our box head directly outputs the absolute bounding box representation itself, as described above in Appendix F.2.1.

We use the same training method and optimizer as described in appendix A1.3 in (Matthias Minderer, 2022), replacing the gIoU weight with a weight for our 8 corner vertex loss described above in Appendix F.2.2 but keeping the same values. To improve training speed, we randomly subsample 12 objects from each image for each iteration. However, during evaluation we include all objects in the scene when computing the 3D IOU. We train with a batch size of 1,024 for 250k iterations with an initial learning rate of $1e-4$.

### F.4. Results From All Runs

|      | Baseline | RoPE | RoPE-M | Circulant-S | Cayley-S |
|------|----------|------|--------|-------------|----------|
|      | **49.77** | 55.77 | 2.53 | 56.69 | **58.85** |
| ViT  | 49.10 | 52.11 | **57.17** | **58.95** | 58.43 |
|      | 47.85 | **58.09** | 2.46 | 13.12 | 57.47 |
|      | 65.88 | **71.21** | 70.78 | **72.36** | 70.36 |
| ViTD | 66.49 | 69.50 | 69.94 | 69.86 | **72.67** |
|      | **67.60** | 70.31 | **70.90** | 68.51 | 71.91 |

*Table 7.* Average 3D IOU % over all objects for the 3D bounding box prediction task. For each configuration, 3 models were trained with different random seeds. Baseline indicates no RoPE or STRING. The maximum for each configuration is highlighted in **bold**. Higher is better.

For each configuration of RoPE/STRING and ViT/ViTD, we trained 3 models with different random seeds. In Table 5 in Section 4.2.2 we report the maximum IOU of the 3 for each configuration. Here, Table 7 shows the IOU for every run, with the maximum highlighted in bold. Note that for 3 runs (2 for ViT+RoPE-M and 1 for ViT+Circulant-STRING), the models fell into a local minima which they never left and thus had inferior performance. See Appendix F.2.4 for details on possible causes.

## G. Details of Generative Robotics Policies

Figure 14 shows the network architecture for the generative robotics policies for manipulation tasks. PaliGemma (Beyer et al., 2024) VLM with embedding size 256 and patch size 16 is used for image encoding. The policy is trained with Adam optimizer with $1e-4$ learning rate and $1e-4$ weight decay. We use a linear learning rate warm-up for first 10000 steps of training. The policy is trained for a total of 500000 steps with batch size 256.

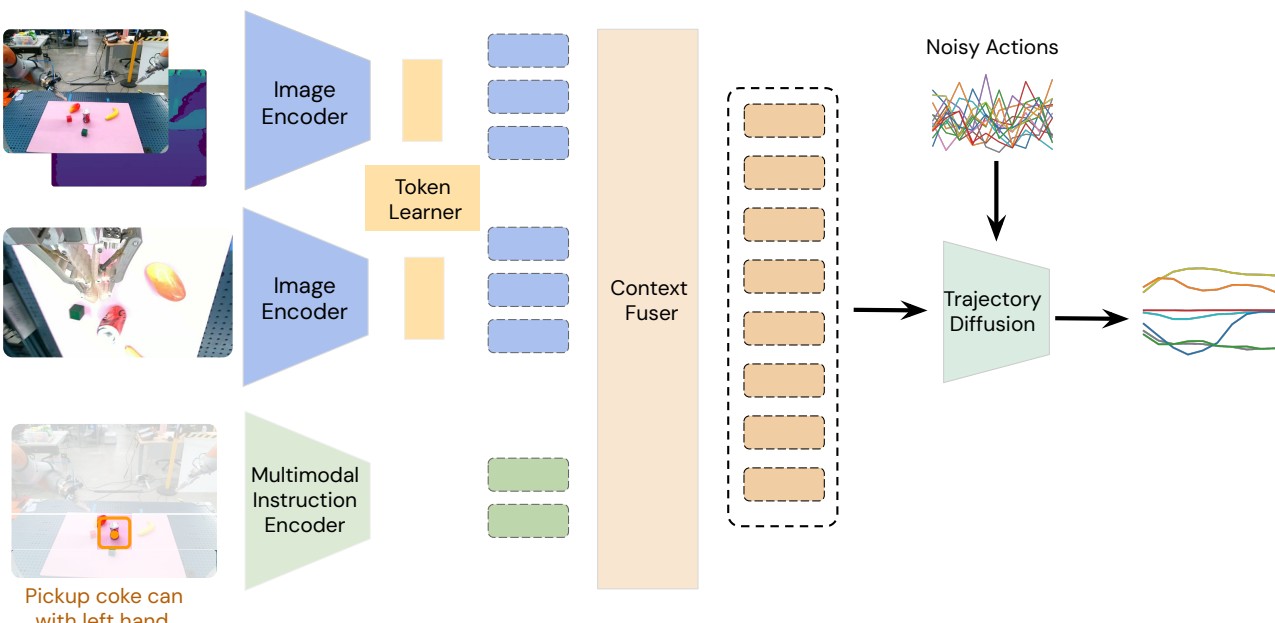

*Figure 14.* Generative policy architecture for robotic manipulation using Kuka-IIWA arms.

