# OpenReview forum: "Learning the RoPEs: Better 2D and 3D Position Encodings with STRING"
_ICML.cc/2025/Conference — ICML 2025 spotlightposter_

### Official Review · Reviewer_PFCP · 2025-03-10

**Overall Recommendation:** 3

**Summary:**

This submission proposes STRING, a novel method that generalizes the popular Rotary Position Encodings (RoPE) used in Transformers. Unlike RoPE, which is naturally suited for 1D inputs and then extended to higher dimensions, STRING offers a theoretical framework that accommodates multi-dimensional (2D/3D) inputs while preserving essential properties like translational invariance and separability. Through extensive experiments—covering image classification, open-vocabulary detection, simulation-based robotics tasks, and real-world robot manipulation—STRING consistently outperforms or matches RoPE and standard baselines. The authors also provide in-depth theoretical justification showing that STRING is, in fact, the most general family of rotation-based position encodings (under certain smoothness constraints).

## After Rebuttal
The authors' reply not well addressed my concerns on 3D imitation learning part. I might keep my score as weak accept.

**Claims And Evidence:**

Yes

**Essential References Not Discussed:**

N/A

**Experimental Designs Or Analyses:**

The experiments cover several aspects, including vision classification and retrieval, object detection, and robotic manipulation.
The authors even conduct the real-world 3D robot manipulation to show the real world effectiveness of String.
However, for the real-world 3D Robot Manipulation part, the authors only show that 3D String is better than 2D, which is actually widely shown [1, 2]. Could authors try some simple 3D-based imitation learning methods [1,2] and showcase whether String could directly improve them?

[1] Ze et al. "3d diffusion policy: Generalizable visuomotor policy learning via simple 3d representations." arXiv preprint arXiv:2403.03954 (2024).
[2] Ke e tal. "3d diffuser actor: Policy diffusion with 3d scene representations." arXiv preprint arXiv:2402.10885 (2024).

**Methods And Evaluation Criteria:**

Yes

**Other Comments Or Suggestions:**

N/A

**Other Strengths And Weaknesses:**

1. The introduction is well written, clearly introducing the context and the motivation.
2. The experiments covering several domains are very extensive.

**Questions For Authors:**

As mentioned above, for the 3D robot manipulation part, it will be more convincing to showcase how STRING improve 3D-based imitation learning methods instead of 2D baselines.

**Relation To Broader Scientific Literature:**

The proposed method generalizes RoPE with a unified rhetorical framework, which is related and useful to the community.

**Theoretical Claims:**

Yes

---

> ### Author Rebuttal · Authors · 2025-04-01
>
> We would like to sincerely thank the Reviewer for the comments. We provide detailed answers below.
>
> $\textbf{Additional tests, e.g. 3D-based imitation learning methods from [1,2]}$:
>
> Thank you very much for the comment. The main goal of the 3D bi-arm KUKA experiments was to combine the best techniques to obtain new SOTA. We have already established STRING as providing gains on the top of the regular 3D algorithms, that in turn improved upon their 2D counterparts in Sec. 4.2.2., where we focused on 3D object localization (see: Table 4). We explicitly describe this goal, putting our plan for bi-arm KUKA experiments in the context of our previous findings,  at the beginning of Sec. 4.4 (l.372-375: $\textit{“Establishing STRING as superior to other methods on previous tasks, ...”}$).
>
> Having said that, we still included in the bi-arm KUKA section 3D-based imitation learning methods that do not leverage STRING. We trained via imitation learning a policy directly using depth to produce normal maps (paragraph: $\textit{“Implicit depth via normal maps”}$ at the beginning of Sec. 4.4.2). This is based on well established methods for incorporating depth into robotics tasks from [1]. As we show in Fig. 5, STRING provides ~$\textbf{11}$% improvement accuracy-wise, as compared to that baseline.
>
> We would like to note one more thing. The 2D policies we compared against in the bi-arm KUKA section were strong enough to outperform various 3D counterparts given similar computational budget (e.g. point-cloud-based methods that were used in the first paper pointed by the Reviewer; we will explicitly clarify it in the camera-ready version). This was the case since point cloud (PC) approaches suffer from the very high computational footprint. The average number of points in the bi-arm KUKA PCs was of order 5K+. Training/deploying policies directly using all the points was infeasible and various tokenization techniques to reduce the number of 3D tokens had a detrimental effect on the performance. Furthermore, base 2D policies applied in that section used vision encoders pre-trained on massive amount of vision data. Trained-from-scratch 3D encoders were not a match for them. Thus it was important for us to build on the 2D policies in that section. The approach we used there with the positional encoding mechanism in the form of STRING applied on RGB-D images via the 3D patch-lifting (see paragraph: $\textit{“Lifting patches to 3D for STRING”}$, p.7) enabled us to successfully do it. First of all: it did not require working with PCs and retained the same number of patches as its 2D counterpart. That was critical for computational efficiency. Secondly, it could easily reuse 2D-pretrained checkpoints, leveraging vision input pre-processing from the 2D baseline as is, only to add depth signal to modulate attention computation. The STRING variant re-used all pre-trained parameters of its 2D counterpart, introducing a negligible amount of additional 3D-specific trainable parameters. Finally, STRING variant could be set up in such a way that at the very beginning of its training it was equivalent to the 2D variant. This was achieved by setting up a depth scaling parameter for the patch-depth (obtained by averaging over depth values of its constituting pixels) as zero at the beginning of training. We commented on some of these in the paper (e.g. a paragraph “From 2D to 3D with STRING”, p. 8), but will provide more detailed discussion in the camera-ready version.
>
> We also would like to note that we added several additional experiments for the rebuttal, confirming the effectiveness of STRING:
>
> 1. We re-ran the evaluations for the $\textit{BowlOnRack}$ task (we chose one task, given rebuttal period time constraints) from the Aloha-sim portfolio, by increasing the number of trials $\textbf{from 10 to 100}$. The ranking of methods in that setting stays exactly the same as reported before with STRING outperforming others.
> 2. We run experiments for 2D and 3D object detection using the SoViT-400m models which surpass ViT-H and are competitive in performance with ViT-G. As compared to baseline, STRING provides >$\textbf{4.7}$% relative improvement on COCO AP and >$\textbf{5.1}$% relative improvement on LVIS AP and improves also upon regular RoPE in the OWL-ViT-2D setting. For 3D object detection with SoViT-400m models, using STRING led to additional accuracy gains as compared to the baseline ($\textbf{1.2}$ of the baseline improvement coming from using a larger model).
> 3. We also conducted additional scaling experiments, where we increased the image resolution from 256 x 256 to 512 x 512. STRING maintained an advantage over the baseline for the object detection task from Sec. 4.2.2 (3D bounding boxes): $\textbf{6}$%+ for the 2D variant and almost $\textbf{10}$% for the 3D variant.
>
>
> $\underline{References}$:
>
> [1] Early or late fusion matters: Efficient rgb-d fusion in vision transformers for 3d object recognition; Tziafas et al.; IROS 2023.

---

### Official Review · Reviewer_1d62 · 2025-03-14

**Overall Recommendation:** 4

**Summary:**

This paper proposes STRING, which generalizes the 2D rotation matrix in position encoding in RoPEs to a more general form of rotation, which is parameterized by the linear combinations of skew-symmetric generator matrices. This allows the framework to obtain exact translational invariant or rotational invariant, depending on the needs. The proposed method is tested on 2D and 3D object detection tasks, as well as a robot policy learning task, in which the proposed method is shown to obtain better performance than ViT and RoPEs.

**Claims And Evidence:**

The main contribution of this paper is to generalize the position encoding. Another contribution is that this position encoding can be extended to 3D. The authors provide a detailed explanation of their position encoding design and proof of its invariance.

**Essential References Not Discussed:**

The references are addressed adequately.

**Experimental Designs Or Analyses:**

The authors provide extensive experiments on 2D object detection, 3D object detection, and policy learning for robot manipulation. All of them are compared with standard baselines ViT and RoPE. While the improvements are marginal, this is the nature of generalizing RoPE. The out-of-distribution evaluation shows the robustness if the framework is extended to 3D.
Additional discussion on the differences between Cayley-STRING and Circulant-STRING can strengthen the paper.

**Methods And Evaluation Criteria:**

Evaluation metrics are standard. There are two ways to learn the skew-symmetric matrix: Cayley-STRING and Circulant-STRING. While they are each introduced clearly, a discussion about the trade-off and when to use which would be beneficial.

**Other Comments Or Suggestions:**

N/A

**Other Strengths And Weaknesses:**

N/A

**Questions For Authors:**

How does the discretization in the sensory data (for example, limited image resolution/token) affect the performance of the system?

**Relation To Broader Scientific Literature:**

Overall, the paper is a great addition to the literature. It generalizes RoPEs by a natural choice of position encodings. The paper is both theoretically sound and well supported by extensive experimental results.

**Theoretical Claims:**

The paper provides theorems of the position encoding based on the generators, which proves how STRING is a generalization of RoPE and the translational invariant property.

---

> ### Author Rebuttal · Authors · 2025-04-01
>
> We would like to sincerely thank the Reviewer for the comments. We provide detailed answers below.
>
> $\textbf{Discussion on trade-offs regarding using Cayley-STRING and Circulant-STRING}$:
>
> Thank you very much for an excellent comment. With token dimensionality per head $d_{\mathrm{head}}$, Cayley-STRING introduces $d_{\mathrm{head}}(d_{\mathrm{head}} - 1) / 2+ d_{\mathrm{head}} / 2$ parameters per head while Circulant-STRING adds $d_{\mathrm{head}}$ parameters per head. This is the case since Cayley-STRING trains a full skew-symmetric matrix and d / 2 frequencies, while Circulant-STRING uses a learnable circulant matrix which is fully determined by its first row. Furthermore, multiplication with Cayley-STRING matrices takes quadratic time in $d_{head}$ during training, whereas multiplication with Circulant-STRING matrices takes time $O(d_{\mathrm{head}} \log(d_{\mathrm{head}}))$ via Fast Fourier Transform (FFT). During inference, quadratic cost can be in principle absorbed into existing q/k projections at no extra cost (for the vanilla attention mechanism). Both approaches improve upon regular RoPE, yet Cayley-STRING in general leads to larger improvements. Thus we have here a classic quality-speed tradeoff. For applications with strict memory constraints, we recommend to use Circulant-STRING due to its very compact computational footprint, whereas for other applications Cayley-STRING is recommended. We will add the above discussion to the camera-ready version of the paper.
>
> ===========================================================================================================================================================
>
> $\textbf{The impact of discretization in the sensory data on the performance of the system}:$
>
> Thank you for your question. We conducted our experiments with images of various resolutions, seeing consistent gains across the board, as compared to the baselines. The statement that STRING is effective even if the sensory data is not accurate (due to discretization or noise) is also supported by our 3D experiments with the bi-arm KUKA robot. In all those experiments, depth sensors provide a relatively noisy signal, yet STRING is capable of leveraging depth sensors to provide $\textbf{8}$%+ accuracy improvements in regular in-distribution tasks and from $\textbf{15}$% to $\textbf{42}$% improvements for out-of-distribution tasks. We will add this discussion to the camera-ready version of the paper.

---

### Official Review · Reviewer_CQUV · 2025-03-16

**Overall Recommendation:** 4

**Summary:**

This paper proposes a class of positional encoding for high-dimensional tokens with both separability and translation invariance. The proposed positional encoding is a generalization to the rotary positional encodings (RoPE) based on Lie groups. The authors also provide a computationally efficient implementation of it. Experiments show its gains with transformer models in robotics and vision tasks.

## update after rebuttal
Thank the authors for their responses! My questions are well-answered.

**Claims And Evidence:**

The three key contributions claimed in the paper are:
- "We introduce STRING, a new family of position encodings for multidimensional token coordinates that respect both separability and translational invariance."
	- The proposed method is indeed a "family" of positional encodings with different variants.
	- Both separability and translation invariance are well discussed in the paper.
- "We rigorously analyse STRING’s theoretical properties ( Sec. 3), proving that it is more general than RoPE. We provide computationally efficient implementations."
	- The theoretical properties are well provided.
	- The relations between STRING and RoPE are well discussed.
	- For computational efficiency, there are some theoretical discussions about computational complexity, but it would be good to also have some statistics or analysis on its time or memory consumption in the experiments.
- "We show strong accuracy gains across varied models using Transformers with STRING, on a range of robotics and general vision tasks (see Fig. 1 and Sec. 4)."
	- The experimental results support this argument.

**Essential References Not Discussed:**

The related works and preliminaries have provided a good background for the topic studied in this paper.

**Experimental Designs Or Analyses:**

The experimental designs look reasonable to me, especially the focus on comparing STRING (with its variants) to previous positional encodings. The Training curve in Fig. 4 also looks good, as it shows that networks with STRING positional encodings are also easy to optimize for.

It may be good to also have some statistics or discussions on the number of parameters, memory consumption, or training/inference time, as efficient implementation is discussed in the key contributions.

**Methods And Evaluation Criteria:**

The method and evaluations look reasonable to me. The evaluations are mostly done on public benchmarks with widely adopted metrics.

**Other Comments Or Suggestions:**

- It may be good to have clearer explanations of the differences between Theorem 3.3 and 3.4, instead of calling both of them "RoPE is a type of STRING". Based on my understanding, Theorem 3.3 is saying that RoPE is a special case of STRING (and thus STRING is a generalization of RoPE), and Theorem 3.4 is about, and Theorem 3.4 is saying that all STRING positional encodings can be expressed with RoPE.
- It may be good to have a clearer introduction to the notations at the beginning. For example, it took me some time to find what are $d_c$ and $d$.

**Other Strengths And Weaknesses:**

To summarize, the strenghts and weaknesses are:

Strengths:
- A general framework with theoretical proofs, well-designed details, and different variants.
- Good experimental results.

Weaknesses:
- Computational efficiency: This point can be better justified if there could be some statistics or discussions on the number of parameters, memory consumption, or training/inference time.
- More explanations on Theorem 3.3 and 3.4 and the notations could make the paper easier to read.

**Questions For Authors:**

I don't have specific questions in addition to the comments I made above.

**Relation To Broader Scientific Literature:**

My personal feeling is that the proposed method is a generalization of RoPE (and thus RoPE is its closest related work). But this generalization is non-trivial -- it covers the most general cases and shows different variants. Computational efficiency is also studied in this generalization. So overall I think the proposed method has good novelty and well-developed details.

**Theoretical Claims:**

I briefly went through the proofs for Theorem 3.2-3.4. Theorem 3.2 is mostly based on the general conclusions from group theory and I think they are good. The intuitions for Theorem 3.3 and 3.4 look right, where I briefly followed the proofs but didn't check all computations in details.

For Theorem 3.5, I am not very familiar with the computational complexity of FFT and DFT algorithms, and thus I cannot say too much about the proof. But a small question I have is, it seems that this complexity analysis is mostly about the forward computation? How would the complexity be like for backpropogation?

---

> ### Author Rebuttal · Authors · 2025-04-01
>
> We would like to sincerely thank the Reviewer for the comments. We provide detailed answers below.
>
> $\textbf{Theorem 3.5}$:
>
> The proof relies on the fact that the computation of $exp(\mathbf{M})$ for a given matrix $\mathbf{M}$ can be re-written as: $\exp(\mathbf{M}) = \Sigma \exp(\mathbf{D}) \Sigma^{-1}$ for a given factorization:
> $\mathbf{M}=\Sigma \mathbf{D} \Sigma^{-1}$ (from the Taylor-series definition of matrix exponentiation). The next observation is that, for the matrices of interest in the proof, the factorization can be done efficiently with $\Sigma$ and $\mathbf{D}$ being: (1) a DFT matrix and (2) a diagonal matrix with the main diagonal given as the action of $\Sigma$ on an easy-to-compute vector, respectively. The final observation is that the DFT matrix supports fast matrix-vector multiplication (via the Fast Fourier Transform) and the diagonal matrix exponentiation is obtained by element-wise exponentiation. We will clarify it in the final version of the paper by providing a sketch of the proof in the main body of the paper.
>
> Thank you very much for an excellent question regarding the backpropagation pass. Even though the analysis in the paper was made for the forward pass, it can be extended to the backward pass. This comes from the fact that the computation of the gradient in the backpropagation over networks leveraging Circulant-STRING matrices also involves multiplications with DFT matrices. The analysis of the backpropagation is more technical, but basically follows steps from Sec. 4.1 of [1]. We will provide a paragraph to discuss computational gains in backpropagation in the camera-ready version of the paper.
>
> $\underline{References}$:
>
> [1]  An Exploration of Parameter Redundancy in Deep Networks with Circulant Projections; Cheng et al.; ICCV 2015.
>
> ===========================================================================================================================================================
>
> $\textbf{Statistics or discussions on the number of parameters, memory consumption, or training/inference time}$:
>
> Thank you very much for the comment. STRING introduces a negligible amount of extra trainable parameters. With token dimensionality per head $d_{\mathrm{head}}$, Cayley-STRING introduces $d_{\mathrm{head}}(d_{\mathrm{head}} - 1) / 2 + d_{\mathrm{head}} / 2$ parameters per head while Circulant-STRING adds $d_{\mathrm{head}}$ parameters per head.
> Thus for instance for the: (1) Cayley-STRING variant with no parameter sharing across heads, (2) Cayley-STRING with parameters shared across heads and (3) Circulant-STRING with no parameter sharing across heads, parameter count increase is marginal: (1) 0.34%, (2) 0.028% and (3) 0.016% respectively. In terms of training time and memory footprint, the performance of Cayley-STRING and Circulant-STRING is similar to that of the RoPE-M model (~$\pm 7 \$%). We will include these numbers in the supplementary material of the camera-ready version of the paper.
>
> ===========================================================================================================================================================
>
> $\textbf{Theorem 3.3 and 3.4}$:
>
> We thank the Reviewer for the comment, and will ensure that our notation is consistent and clear. Theorem 3.3 shows that RoPE is a special case of STRING for a particular choice of an antisymmetric generator. Meanwhile, Theorem 3.4 shows that RoPE is a special case of STRING with basis transformation matrix P = I. Both theoretical results show that our new method is more general than that previous SOTA, but from complementary perspectives. As shown in the paper, this increased generality unlocks better performance in experiments. We will make sure this is clear in the text and highlight the differences between these important, closely-related results.
>
> ===========================================================================================================================================================
>
> $\textbf{Clearer introduction to the notation at the beginning}$:
>
> Thank you for the question. $d$ refers to the dimensionality of the latent representations in the Transformer, i.e. the length of queries, keys and values. Meanwhile, $d_c$ refers to the dimensionality of the position coordinates encoded using STRING, i.e. 1 for sequences (text), 2 for 2D data (images), and 3 for 3D data (images with depth). We will make sure this is clear in the text.

---

### Official Review · Reviewer_57Ag · 2025-03-21

**Overall Recommendation:** 3

**Summary:**

This paper introduces STRING (Separable Translationally Invariant Position Encodings), a new family of position encodings. STRING extends RoPE using a theoretical framework based on Lie groups, maintaining key properties like separability and translational invariance, while supporting 2D/3D token representations.

The authors propose effective implementations, such as Cayley-STRING and Circulant-STRING, and demonstrate through experiments that STRING outperforms RoPE and absolute position encodings (APEs) across tasks.

The experiments include:

Image classification and retrieval.
Open-vocabulary 2D/3D object detection.
Robotic manipulation tasks in both simulation and real-world settings.

**Claims And Evidence:**

The paper’s claims are generally well-supported by clear evidence:


STRING generalizes RoPE with separability and translational invariance.
Supported by rigorous theoretical proofs (Theorem 3.2 and 3.3).

STRING outperforms RoPE and APEs in tasks like classification, detection, and robotics.
Backed by experimental results showing consistent improvements, e.g., higher accuracy in ImageNet and better IOU in 3D object detection.

STRING is computationally efficient.
Supported by proposed implementations (Cayley-STRING, Circulant-STRING) and complexity analysis.

**Essential References Not Discussed:**

N/A

**Experimental Designs Or Analyses:**

Please see the Methods And Evaluation Criteria.

The ALOHA simulation tasks (10 trials per task) and KUKA real-world experiments (50 trials) have limited evaluations, which may not fully capture variability or ensure robustness. Larger-scale testing is needed to strengthen the conclusions.

**Methods And Evaluation Criteria:**

The proposed methods and evaluation criteria are appropriate:

Methods: STRING is well-designed for tasks requiring translationally invariant position encodings, with practical implementations like Cayley-STRING and Circulant-STRING ensuring efficiency.
Evaluation: Benchmarks like ImageNet, WebLI-3D, and robotics tasks (ALOHA and KUKA) are relevant and suitable for testing STRING's performance.

However, the ALOHA simulation tasks (10 trials per task) and KUKA real-world experiments (50 trials) have limited evaluations, which may not fully capture variability or ensure robustness. Larger-scale testing is needed to strengthen the conclusions.

**Other Comments Or Suggestions:**

N/A

**Other Strengths And Weaknesses:**

Strengths:

The paper focuses on the important area of position encoding, which is highly relevant to the broader field of machine learning. It introduces novel theoretical derivations and technical contributions, which are innovative and valuable.

Weaknesses:

**Limited Scalability Experiments:** The paper lacks scaling experiments. It is unclear if Circulant-S and Cayley-S perform well on larger models, such as ViT-H or ViT-G.

**Marginal Improvements Over Baselines:** I noticed that Circulant-S and Cayley-S do not show significant improvements over baselines like RoPE, especially in the ALOHA simulation task, where the results are nearly identical across many tasks.

**Robotics Task Evaluation:** The evaluation on robotics tasks could be more robust. For example, testing on more trials and tasks would strengthen the claims.

While the theoretical contributions are clear, I am not deeply familiar with this field, which limits my ability to assess the novelty of the contributions in the broader context of existing research.

**Questions For Authors:**

I give a Weak Accept because of the solid theoretical proofs and novel technical contributions. However, I am not very familiar with this field, so I cannot guarantee my judgment with full confidence. I did not give a higher score because the experimental results show limited improvements over the baselines, especially due to the lack of scaling-related experiments, which raises concerns about the actual effectiveness of the proposed methods.

**Relation To Broader Scientific Literature:**

The paper builds on prior work in position encoding and equivariant representations, extending these ideas with the STRING framework. It specifically addresses limitations of fixed encodings in transformers and draws inspiration from group theory and circulant structures, contributing a novel approach aligned with recent advances in efficient, invariant neural architectures.

**Theoretical Claims:**

The theoretical claims, including Theorem 3.2 and 3.3, were reviewed and appear to be correct, with clear proofs provided in the appendix.

---

> ### Author Rebuttal · Authors · 2025-04-01
>
> We would like to sincerely thank the Reviewer for the comments. We provide detailed answers below.
>
> $\textbf{Increasing the number of trials in testing (currently 10 trials / task for Aloha-sim and 50 trials for bi-arm KUKA)}:$
>
> We sincerely thank the Reviewer for the comment. We would like to note that 10 trials / task is considered normal for the evaluation of the robotics controllers, both in sim and on hardware (e.g. [1]: 5 repetitions as stated in Sec 6.1, $\textit{"This is repeated \textbf{5} times"}$  and [2]: 10 repetitions in $\textit{“We ran each of the learned controller \textbf{10} times.”}$ at the beginning of Sec. B (“Results and Analysis”)).
>
> In fact it is also considered normal for testing several regular machine learning algorithms. For instance, most papers evaluating Random Fourier Features and related methods, such as [3] ($\textit{“All experiments are repeated \textbf{ten} times”}$, p.7), [4] (caption of Fig. 2 $\textit{“\textbf{10} independent trials are executed”}$) that usually can be quickly evaluated with very limited computational resources, apply 10 repetitions to calculate empirical MSEs.
>
> Having said that, following Reviewer’s comment for the rebuttal, we re-ran the evaluations for the $\textit{BowlOnRack}$ task (we chose one task, given rebuttal period time constraints) from the Aloha simulation portfolio, by increasing the number of trials $\textbf{from 10 to 100}$. The ranking of different methods in that setting $\textbf{stays exactly the same}$ as reported before: 1. STRING (83% accuracy) 2. regular ViT baseline (80% accuracy) and 3. RoPE (75%). We also would like to emphasize that all reported KUKA experiments are $\textbf{on-robot}$ (and thus much more time-consuming) and in the paper we use many more trials for the corresponding evals.
>
> $\underline{References}$:
>
> [1] Scalable Deep Reinforcement Learning for Vision-Based Robotic Manipulation; Kalashnikov et al.; CoRL 2018; $\textbf{Best Systems Paper, ~1.8K citations}$.
>
> [2] Deep Spatial Autoencoders for Visuomotor Learning; Finn et al., ICRA 2016; $\textbf{764 citations}$.
>
> [3] Nystrom Method vs Random Fourier Features: A Theoretical and Empirical Comparison; Yang et al.; NeurIPS 2012; 450+ citations).
>
> [4] Quasi-Monte Carlo Feature Maps for Shift-Invariant Kernels; Avron et al.; ICML 2014
>
> ===========================================================================================================================================================
>
> $\textbf{Scaling Experiments to show that STRING still performs well}:$
>
> Thank you very much for an interesting question about scaling to larger models. Following the Reviewer’s suggestion, we have run such experiments using the SoViT-400m class of models which surpass ViT-H and are competitive in performance with ViT-G [1]. As compared to baseline, STRING provides $\textbf{>4.7}$% relative improvement on COCO AP and $\textbf{>5.1}$% relative improvement on LVIS AP and improves also upon regular RoPE in the OWL-ViT-2D setting. For the task of 3D object detection with SoViT-400m models, using STRING led to additional accuracy gains as compared to the baseline ($\textbf{1.2}$ of the baseline improvement coming from using a larger model).
> Thus, we tested that STRING does perform well also on larger models.
>
> We have also conducted additional scaling experiments, where we increased the resolution of images from 256 x 256 to 512 x 512. In the higher-resolution setting, STRING maintained an advantage over the baseline for the object detection task from Sec. 4.2.2 (3D bounding boxes): $\textbf{6}$%+ for the 2D variant and almost $\textbf{10}$% for the 3D variant.
>
> We will include all these results in the camera-ready version of the paper.
>
> $\underline{References}$:
>
> [1] Getting ViT in Shape: Scaling Laws for Compute-Optimal Model Design; Alabdulmohsin et al.; NeurIPS 2023
>
> ===========================================================================================================================================================
>
> $\textbf{Improvements Over Baselines}$:
>
> Thank you very much for your question. The STRING variant provides highest average accuracy over all Aloha-sim tasks. However, what is as important and clearly seen in Fig. 4, it also provides significant convergence improvements in training, achieving accuracy level of regular RoPE in $\textbf{1.6x}$ shorter time. We explicitly commented on that in l.367-368. This clearly distinguished STRING from regular RoPE. Furthermore, our 3D object detection experiments clearly show improvements coming from STRING (from approximately $\textbf{1}$% to $\textbf{1.5}$%), in both: 2D and 3D settings, as compared to $\textbf{the best}$ RoPE variants (Table 4). These improvements are actually visible to the naked eye. We show it in Fig. 2, where predicted small 3D boxes (in blue) are clearly misaligned with ground truth bounding boxes (in green) for baseline and RoPE, while pretty well aligned for STRING.

---

> > ### Comment · Reviewer_57Ag · 2025-04-08
> >
> > Thank the authors for answering my question. I will maintain my original positive score.

---

### Decision · Program_Chairs · 2025-05-01

**Decision:**

Accept (spotlight poster)

**Comment:**

In this paper, the authors introduce STRING (Separable Translationally Invariant Position Encodings), a novel family of position encodings for Transformer models. STRING generalizes the widely used Rotary Position Encoding (RoPE) using a Lie group theoretical framework. This allows it to natively handle multi-dimensional inputs (1D/2D/3D) while preserving key properties like separability and translational invariance, and makes it especially relevant for robotics/computer vision. The authors propose efficient implementations (Cayley-STRING, Circulant-STRING) and demonstrate performance improvements over RoPE and absolute position encodings across diverse tasks, including image classification/retrieval, 2D/3D object detection, and simulated/real-world robotic manipulation.

The reviewes were generally positive, with reviewers particularly highlighting the theoretical soundness of the proposed method, novelty, sufficiency of experimental evaluation, as well as the well written nature of the paper. Some concerns were also raised about the number of trials in robotics experiments and the lack of scaling experiments to larger models. These were largely addressed in the rebuttal with additional experiments (larger SoViT models, higher resolution, more trials for one sim task). The authors also addressed other reviewer asks such as runtime performance/memory numbers; tradeoffs between Cayley-STRING and Circulant-STRING, which will be added to the final version of the paper.

Though one additional concern from reviewer PFCP was that the authors only show that using STRING, methods that use RGBD outperform the method that used RGB - but this does not necessarily show how well STRING handles native 3D representations. The authors respond that the intent was to practically show that the best performance by adding 3D info efficiently to very strong pre-trained 2D models, and direct 3D models (such as those that operate on point clouds) are more expensive to train and experiment with. The AC agrees with this, and also feels that in light of the other evidence in support of the method, as well as the depth-normal map experiment the authors performed as a 3D baseline, the efficacy of the method is indeed being shown.

Overall, given the strengths of the paper, the theoretically sound contribution, and positive consensus from all the reviewers, I recommend the paper be accepted,